# *Anopheles* salivary antigens as serological biomarkers of vector exposure and malaria transmission: A systematic review with multilevel modelling

Ellen A Kearney[1,2], Paul A Agius[1,2,3], Victor Chaumeau[4,5], Julia C Cutts[1,6], Julie A Simpson[2], Freya JI Fowkes[1,2,3]*

[1]The McFarlane Burnet Institute of Medical Research and Public Health, Melbourne, Australia; [2]Centre for Epidemiology and Biostatistics, Melbourne School of Population and Global Health, The University of Melbourne, Melbourne, Australia; [3]Department of Epidemiology and Preventive Medicine, Monash University, Melbourne, Australia; [4]Shoklo Malaria Research Unit, Mahidol-Oxford Tropical Medicine Research Unit, Faculty of Tropical Medicine, Mahidol University, Mae Sot, Thailand; [5]Centre for Tropical Medicine and Global Health, Nuffield Department of Medicine, University of Oxford, Oxford, United Kingdom; [6]Department of Medicine at the Doherty Institute, The University of Melbourne, Melbourne, Australia

*For correspondence:
fowkes@burnet.edu.au

Competing interest: The authors declare that no competing interests exist.

## Abstract

**Background:** Entomological surveillance for malaria is inherently resource-intensive and produces crude population-level measures of vector exposure which are insensitive in low-transmission settings. Antibodies against *Anopheles* salivary proteins measured at the individual level may serve as proxy biomarkers for vector exposure and malaria transmission, but their relationship is yet to be quantified.

**Methods:** A systematic review of studies measuring antibodies against *Anopheles* salivary antigens (PROSPERO: CRD42020185449). Multilevel modelling (to account for multiple study-specific observations [level 1], nested within study [level 2], and study nested within country [level 3]) estimated associations between seroprevalence with *Anopheles* human biting rate (HBR) and malaria transmission measures.

**Results:** From 3981 studies identified in literature searches, 42 studies across 16 countries were included contributing 393 study-specific observations of anti-*Anopheles* salivary antibodies determined in 42,764 samples. A positive association between HBR (log transformed) and seroprevalence was found; overall a twofold (100% relative) increase in HBR was associated with a 23% increase in odds of seropositivity (OR: 1.23, 95% CI: 1.10–1.37; p<0.001). The association between HBR and *Anopheles* salivary antibodies was strongest with concordant, rather than discordant, *Anopheles* species. Seroprevalence was also significantly positively associated with established epidemiological measures of malaria transmission: entomological inoculation rate, *Plasmodium* spp. prevalence, and malarial endemicity class.

**Conclusions:** *Anopheles* salivary antibody biomarkers can serve as a proxy measure for HBR and malaria transmission, and could monitor malaria receptivity of a population to sustain malaria transmission. Validation of *Anopheles* species-specific biomarkers is important given the global heterogeneity in the distribution of *Anopheles* species. Salivary biomarkers have the potential to transform surveillance by replacing impractical, inaccurate entomological investigations, especially in areas progressing towards malaria elimination.

**Funding:** Australian National Health and Medical Research Council, Wellcome Trust.

## Editor's evaluation

We believe this systematic review on the use of serological data to monitor anopheline mosquito exposure will add to the existing literature and help provide important insight into how these markers may be used to understand malaria transmission.

## Introduction

Sensitive and accurate tools to measure and monitor changes in malaria transmission are essential to track progress towards malaria control and elimination goals. Currently, the gold standard measurement of malaria transmission intensity is the entomological inoculation rate (EIR), a population measure defined as the number of infective *Anopheles* mosquito bites a person receives per unit of time. EIR is calculated as the human biting rate (HBR; measured at the population level by entomological vector-sampling methodologies [gold standard: human landing catch]) multiplied by the sporozoite index (proportion of captured *Anopheles* with sporozoites present in their salivary glands). However, estimation of EIR and HBR via entomological investigations is inherently labour and resource intensive, requiring trained collectors, specialised laboratories, and skilled entomologists. Furthermore, these approaches provide a crude population-level estimate of total vector exposure at a particular time and location, precluding investigation of heterogeneity and natural transmission dynamics of individual-level vector–human interactions (*Monroe et al., 2020*). For example, indoor human landing catches provide poor estimates of outdoor biting and thus total vector exposure (*Mathenge et al., 2005*). The sensitivity of EIR is further compromised in low transmission settings where the number of *Plasmodium*-infected specimens detected is low and often zero.

Evaluation of the human antibody response to *Anopheles* spp. salivary proteins has the potential to be a logistically practical approach to estimate levels of exposure to vector bites at an individual level. Several *Anopheles* salivary proteins have been shown to be immunogenic in individuals naturally exposed to the bites of *Anopheles* vectors and have been investigated as serological biomarkers to measure *Anopheles* exposure (*Badu et al., 2012b*; *Drame et al., 2013a*; *Drame et al., 2010a*; *Drame et al., 2010b*; *Drame et al., 2015*; *Rizzo et al., 2011a*; *Rizzo et al., 2011b*; *Stone et al., 2012*; *Drame et al., 2012*), malaria transmission (*Londono-Renteria et al., 2015a*; *Ya-Umphan et al., 2017*; *Noukpo et al., 2016*), and as an outcome for vector control intervention studies (*Drame et al., 2013a*; *Drame et al., 2010a*; *Drame et al., 2010b*; *Noukpo et al., 2016*; *Idris et al., 2017*). However, a major shortcoming of the literature is that studies are largely descriptive and do not quantify the association between entomological and malariometric measures and anti-*Anopheles* salivary antibody responses. We undertook a systematic review with multilevel modelling to quantify the association between HBR, EIR, and other markers of malaria transmission, with anti-*Anopheles* salivary antibody responses, and to understand how these associations vary according to transmission setting and dominant *Anopheles* vectors which can exhibit different biting behaviours. In particular, we were interested in comparing the African context (where *Anopheles gambiae* and *Plasmodium falciparum* predominates) to non-African settings (where *An. gambiae* is absent and where both *P. falciparum* and *Plasmodium vivax* are prevalent). This knowledge is pertinent to advance the use of salivary antibody biomarkers as a vector and malaria transmission serosurveillance tool.

## Methods
### Search strategy and selection criteria

We performed a systematic review with multilevel modelling according to the Meta-analysis of Observational Studies in Epidemiology (MOOSE) and Preferred Reporting Items for Systematic Reviews and Meta-Analyses (PRISMA) guidelines (*Moher et al., 2009*; *Stroup et al., 2000*) (Reporting Standards Document). Five databases were searched for published studies investigating antibodies to *Anopheles* salivary antigens as a biomarker for mosquito exposure or malaria transmission published before 30 June 2020. The protocol (Appendix 1) was registered with PROSPERO (CRD42020185449).

The primary criterion for inclusion in this systematic review was the reporting of estimates of sero-prevalence or total levels of immunoglobulin (Ig) in human sera against *Anopheles* salivary antigens. We considered for inclusion cross-sectional, cohort, intervention, and case–control studies of individuals or populations living in all geographies with natural exposure to *Anopheles* mosquitoes. Studies that were solely performed in participants not representative of the wider naturally exposed population (i.e. mosquito-allergic patients, soldiers, returned travellers) were excluded.

## Measures
### Outcomes
The primary outcome of our systematic review was antibodies (seroprevalence or levels, including all Ig isotypes and subclasses) against any *Anopheles* salivary antigens (full-length recombinant proteins, peptides, and crude salivary extract). Study-reported salivary antibody data was extracted at the most granular level (i.e. for each site; time point), with each observation of seroprevalence or levels included as a study-specific salivary antibody observation. As measurement of antibody levels does not produce a common metric between studies, only values of seroprevalence could be included in multilevel modelling analyses. Therefore, to maximise data, authors of studies that reported only antibody levels were contacted and asked to classify their participants as 'responders' or 'non-responders' according to seropositivity (antibody level relative to unexposed sera). Studies that provided antibody levels or categorised seropositivity based upon arbitrary cut-offs are included in narrative terms only.

### Exposures
The primary exposures of interest were the entomological metrics HBR (average number of bites received per person per night) and EIR (infectious bites received per person per year). Secondary exposures included study-reported prevalence of *Plasmodium* spp. infection (confirmed by either microscopy, rapid diagnostic test (RDT), or polymerase chain reaction [PCR]) and seroprevalence of antimalarial antibodies against pre-erythrocytic and blood stage *Plasmodium* spp. antigens. Where exposure estimates were not provided, we attempted to source data from other publications by the authors or used the site geolocation (longitude and latitude) and year to obtain estimates of EIR from the Pangaea dataset (**Yamba et al., 2018**), *P. falciparum* rates in 2–10 year olds ($Pf$PR$_{2-10}$), and dominant vector species (DVS) from the Malaria Atlas Project (MAP; **The Malaria Atlas, 2017**). Malarial endemicity classes were derived by applying established endemicity cut-offs to MAP $Pf$PR$_{2-10}$ estimates (**Bhatt et al., 2015**). For the purposes of the modelling analyses, we defined DVS as where *An. gambiae* sensu lato (s.l.) was the only DVS, where *An. gambiae* s.l. was present with additional DVS, or where *An. gambiae* s.l. was absent. Studies of salivary antigens where exposure variables could not be sourced and data could not be extracted were excluded.

## Statistical analysis
Where observations of the seroprevalence of antibodies against the same salivary antigen and exposure of interest were reported in more than one study, generalised linear multilevel modelling (mixed effects, logistic) was used to quantify associations between the exposures of interest and salivary antibody seroprevalence measurements (**Song et al., 2019**). Random intercepts for study and country were estimated to account for nested dependencies induced from multiple study-specific salivary antibody observations (level 1) from the same study (level 2) and studies from the same country (level 3). Additionally, study-level random slopes for the entomological and malariometric exposure parameters were estimated to model study-specific heterogeneity in the effect of the exposure of interest (HBR/EIR/malaria prevalence/antimalarial antibody seroprevalence). The associations between the various exposures and the different salivary antigens were analysed separately; however, observations of IgG seroprevalence against the recombinant full-length protein (gSG6) and synthetic peptide (gSG6-P1, the one peptide determined in all studies utilising peptides) form of the gSG6 antigen were analysed together.

Potential effect modification of the associations between exposures and anti-*Anopheles* salivary antibody responses was explored. In analyses quantifying the associations between HBR, as well as EIR, and seropositivity, we included an interaction term with DVS and for vector collection method (human landing catch or other indirect measures, e.g. light traps, spray catches, etc.). For the association between *Plasmodium* spp. prevalence and seropositivity, interaction terms with malaria detection

methodology (light microscopy or PCR) and malarial species (*P. falciparum* only, or *P. falciparum* and *P. vivax*) were estimated.

For the exposure measures (HBR, EIR, malaria prevalence, and antimalarial antibody seroprevalence), the data were log transformed since there were non-linear associations between the exposure measures on the original scale and seroprevalence – supported empirically by superior model fit as indicated by Akaike's information criterion (AIC) and Bayesian information criterion (BIC) fit indices (*Appendix 1—table 1*). To aid interpretation, we present our results as a relative increase in the odds of the gSG6 IgG seropositivity for a twofold or, in other words, a 100% relative increase in the exposures. Intraclass correlation coefficients (ICCs) were estimated for country- and study-specific heterogeneity using estimated model variance components. In order to explore the presence of study-level influence in (HBR and EIR) effect estimate modelling, the Generalised Linear Latent and Mixed Models (gllamm) package (*Rabe-Hesketh et al., 2000*) was used to produce Cooks distance statistics (*Cook, 1977*) at the study level from the generalised linear multilevel models. A conservative cut-off threshold for Cooks distance (4 /*n*) was used to guide sensitivity analyses, where studies were excluded, in turn, to assess outlier influence. All statistical analyses were performed using STATA v15.1.

## Risk of bias in individual studies

Risk of bias was assessed by one reviewer using the Risk of Bias in Prevalence Studies tool (*Hoy et al., 2012*). The risk of bias pertains to the reported observations of anti-*Anopheles* salivary antibody seroprevalence included in the multilevel modelling.

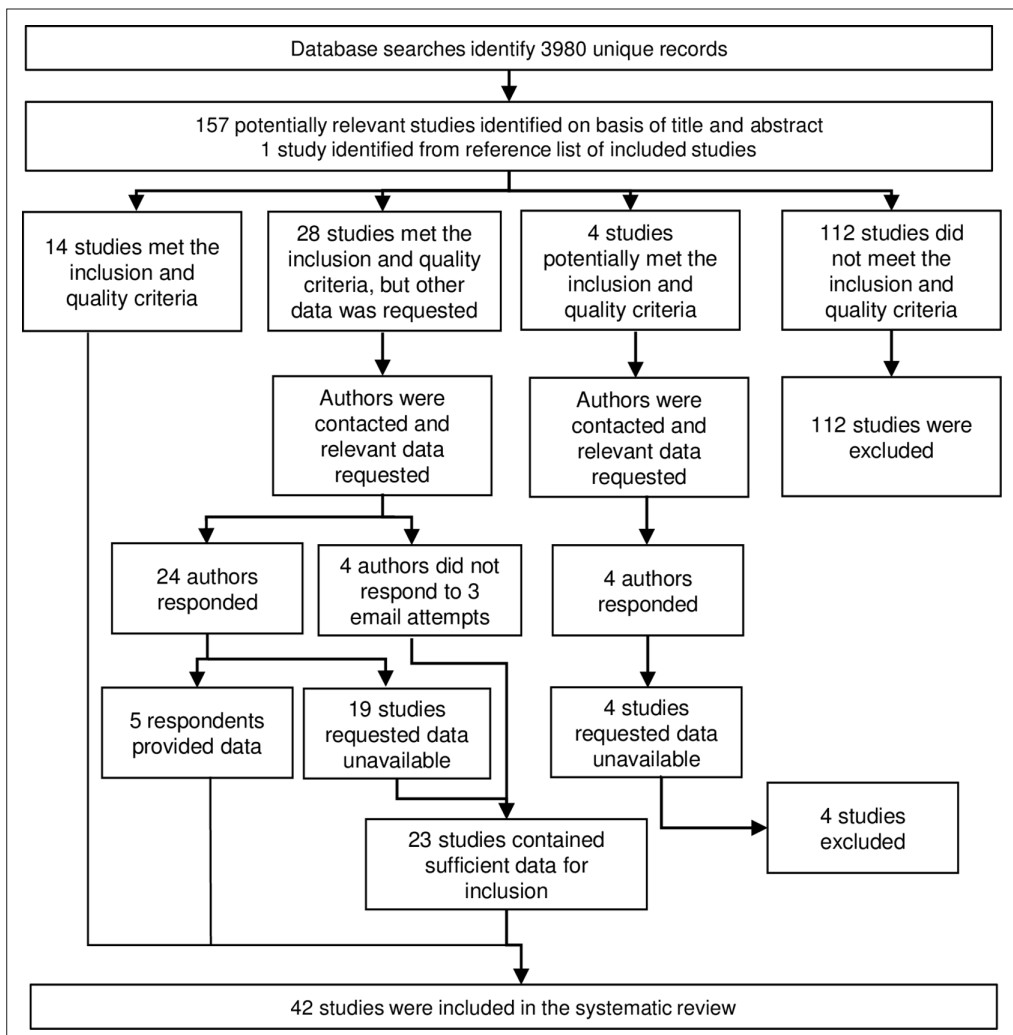

**Figure 1.** Flow diagram of study identification. Excluded studies are detailed in Appendix 3.

# Results

Literature searches identified 158 potentially relevant studies, of which 42 studies were included in the systematic review (*Figure 1*) and are described in *Table 1*. From these studies, we extracted n = 393 study-specific observations of anti-*Anopheles* salivary antibodies determined from antibody measurements in a total of 42,764 sera samples. These studies were performed in 16 countries mostly in hypo- or mesoendemic areas of Africa (32 studies), with a minority performed in South America (four studies), Asia (four studies), and the Pacific (two studies). Studies were classified according to their DVS which reflected the region where the study was conducted. *An. gambiae* s.l. was a DVS in all African study sites (n = 151 study-specific observations from 23 studies where *An. gambiae* s.l. was the only DVS and n = 68 from 16 studies where *An. gambiae* s.l. was present with additional DVS [i.e. *Anopheles funestus*, *Anopheles pharoensis*]), with the exception of one study, which together with the 10 non-African studies contributed n = 174 study-specific estimates where *An. gambiae* s.l. was absent. Most observations came from cross-sectional (n = 191 from 16 studies) or repeated cross-sectional studies (n = 137 from 18 studies), with n = 60 from cohort studies (six studies) and n = 5 from case–control studies (two studies).

The salivary antigen most commonly assessed was *An. gambiae* salivary gland 6 (gSG6), as a full-length protein (n = 67 from 8 studies) and synthetic peptide (*An. gambiae* salivary gland 6 peptide 1 [gSG6-P1]; n = 270 from 24 studies). Additional salivary antigens assessed included *An. gambiae* gSG6-P2 (n = 119 from three studies), recombinant cE5 (n = 15 from two studies), g-5'nuc (n = 3 from one study), and recombinant *An. funestus* fSG6 (n = 6 from two studies) and f-5'nuc (n = 3 from one study). Seven studies measured antibodies to whole salivary gland extracts (SGE) from *An. gambiae* (n = 24 from four studies), *Anopheles darlingi* (n = 5 from two studies), *Anopheles albimanus* (n = 2 from one study), and *Anopheles dirus* (n = 3 from one study), while one study assessed antibodies against synthetic peptides of *An. albimanus* (n = 2) (*Table 1*). All studies investigated total IgG and only five determined an additional isotype or subclass (*Drame et al., 2015*; *Lawaly et al., 2012*; *Rizzo et al., 2014a*; *Rizzo et al., 2014b*; *Waitayakul et al., 2006*). The paucity of studies investigating these latter-mentioned antibody types and *Anopheles* salivary biomarkers precluded extensive multilevel analyses; instead, we present their associations in narrative terms in Appendix 10. Analyses reported below focus on quantifying the relationships between HBR, EIR, and markers of malaria transmission with total IgG to *An. gambiae* gSG6. The distributions of exposure observations were: HBR (n = 197 from 24 studies, median: 3.0 bites per person per night, IQR: 0.9–12.1; range: 0–121.4), EIR (n = 60 from 8 studies, median: 7.3 infectious bites received per person per year, IQR: 0–36.4; range: 0–585.6), and *Plasmodium* spp. prevalence (n = 266 from 22 studies, median: 9.1%; IQR: 4–22%; range: 0–94.6%).

Generalised linear multilevel modelling (mixed effects, logistic) of n = 132 study-specific observations from 12 studies estimated a positive association between *Anopheles* spp.-HBR (log transformed) and seroprevalence of IgG to *An. gambiae* gSG6 salivary antigen (*Drame et al., 2010a*; *Drame et al., 2015*; *Rizzo et al., 2011a*; *Stone et al., 2012*; *Drame et al., 2012*; *Ya-Umphan et al., 2017*; *Soma et al., 2018*; *Traoré et al., 2019*; *Sagna et al., 2013b*; *Sarr et al., 2012*; *Ali et al., 2012*; *Pollard et al., 2019*; *Figure 2*, *Appendix 4—table 1*). As we have log transformed HBR to account for the non-linear relationship between HBR and log odds of gSG6 IgG seropositivity, we have presented estimated odds ratios for different incremental percent increases in HBR (*Figure 2—figure supplement 1*). For example, the magnitude of the association was such that a twofold (100% relative) increase in HBR was associated with a 23% increase (OR: 1.23; 95% CI: 1.10–1.37; p<0.001) in the odds of anti-gSG6 IgG seropositivity (*Figure 2*). Heterogeneity in the effect of HBR on gSG6 across studies was observed (likelihood ratio $\chi^2(1)$ = 109.25, p<0.001); the 95% reference range of study-specific effects for a twofold increase in HBR ranged from a 12% reduction to a 70% increase in odds (OR: 0.88–1.70). There was no evidence that the association between HBR and gSG6 IgG varied according to vector collection method (human landing catch or other indirect methods; p=0.443) or study design (longitudinal cohort or cross-sectional/repeated cross-sectional; p=0.138). Given the global heterogeneity in the distribution of *Anopheles* species, we sought to quantify the extent to which the association between *An. gambiae* gSG6 IgG seropositivity and HBR is moderated by DVS. We observed that the magnitude of the association between *An. gambiae* gSG6 IgG seropositivity and HBR was greatest in African studies where *An. gambiae* s.l. was the only dominant vector (p<0.001, Appendix 5); a twofold increase in HBR was associated with a 37% increase (OR: 1.37; 95% CI: 1.19–1.58; p<0.001) in the

**Table 1.** Key descriptive information from included studies.

*Africa*

| Study year | Country | Malarial endemicity class | Dominant malaria vector species | Study design | No. participants (samples) | Study-specific n | Vector and malariometric variables | Salivary antibody outcomes (seroprevalence [%];[L]levels) |
|---|---|---|---|---|---|---|---|---|
| **Brosseau et al., 2012** | Angola | Hypoendemic; mesoendemic | An. gambiae s.l.; An. funestus | Cross-sectional[‡] | - (1584) | 6 | Plas+[LM], PfPR | gSGE IgG [L] |
| **Drame et al., 2010a** | Angola | Hypoendemic | An. gambiae s.l. | Cohort | 105 (1470) | 12 | HBR; Plas+[LM], PfPR | gSG6-P1 IgG [%; L] |
| **Drame et al., 2010b** | Angola | Hypoendemic | An. gambiae s.l. | Cohort | 109 (1279) | 12 | HBR; Plas+[LM], PfPR | gSGE IgG [L] |
| **Marie et al., 2015** | Angola | Hypoendemic | An. gambiae s.l. | Cohort | 71 (852) | 12 | HBR; PfPR | gcE5 IgG [L] |
| **Drame et al., 2015** | Benin | Hyperendemic | An. gambiae s.l.; An. funestus | Cohort[‡] | 133 (532) | 4 | HBR; PfPR | gSG6-P1 IgG and IgM [%; L] |
| **Rizzo et al., 2011b** | Burkina Faso | Hyperendemic[*] | An. gambiae s.l. | Repeated cross-sectional | - (2066) | 14 | HBR; EIR; Plas+[LMS] | gSG6 IgG [%; L] |
| **Rizzo et al., 2011a** | Burkina Faso | Hyperendemic[*] | An. gambiae s.l. | Repeated cross-sectional | 335 (335) | 3 | HBR | fSG6 IgG [%; L] |
| **Rizzo et al., 2014a** | Burkina Faso | Hyperendemic[*] | An. gambiae s.l. | Repeated cross-sectional | - (359) | 3 | HBR | gcE5 IgG [%; L]; IgG1 and IgG4 [L] |
| **Rizzo et al., 2014b** | Burkina Faso | Hyperendemic[*] | An. gambiae s.l. | Repeated cross-sectional | 270 (270) | 6 | HBR | gSG6 IgG1 and IgG4 [L] |
| **Soma et al., 2018** | Burkina Faso | Mesoendemic | An. gambiae s.l. | Cross-sectional | 1,728 (273) | 6 | HBR; EIR; Plas+[LM], PfPR | gSG6-P1 IgG [%; L] |
| **Koffi et al., 2015** | Cote d'Ivoire | Hypoendemic; mesoendemic | An. gambiae s.l.; An. funestus[†] | Cross-sectional | 94 (94) | 3 | Plas+[LM], Pf-IgG; PfPR | gSG6-P1 IgG [%; L] |
| **Koffi et al., 2017** | Cote d'Ivoire | Hypoendemic | An. gambiae s.l.; An. funestus[‡] | Repeated cross-sectional | 234 (234) | 5 | Pf-IgG; PfPR | gSG6-P1 IgG [%; L] |
| **Traoré et al., 2018** | Cote d'Ivoire | Hypoendemic | An. gambiae s.l. | Repeated cross-sectional[‡] | 89 (178) | 4 | HBR; Plas+[LM], PfPR | gSG6-P1 IgG [L] |
| **Traoré et al., 2019** | Cote d'Ivoire | Hypoendemic | An. gambiae s.l.; An. funestus[†] | Repeated cross-sectional[‡] | - (442) | 6 | HBR; Plas+[LM], PfPR | gSG6-P1 IgG [%; L] |
| **Sadia-Kacou et al., 2019** | Cote d'Ivoire | Mesoendemic | An. gambiae s.l. | Repeated cross-sectional[‡] | 775 (775) | 8 | PfPR | gSG6-P1 IgG [L] |
| **Badu et al., 2015** | Ghana | Mesoendemic | An. gambiae s.l.; An. funestus[†] | Repeated cross-sectional[‡] | 295 (885) | 3 | Plas+[LM], Pf-IgG; PfPR | gSG6-P1 IgG [%; L] |
| **Badu et al., 2012b** | Kenya | Hypoendemic; mesoendemic | An. gambiae s.l. | Repeated cross-sectional | - (1366) | 5 | EIR; Plas+[LMS], PfPR | gSG6-P1 IgG [%; L] |
| **Sagna et al., 2013b** | Senegal | Hypoendemic; mesoendemic | An. gambiae s.l. | Cohort[‡] | 265 (1325) | 25 | HBR; Plas+[LMS], PfPR | gSG6-P1 IgG [%; L] |
| **Drame et al., 2012** | Senegal | Hypoendemic | An. gambiae s.l. | Cross-sectional | 1010 (1010) | 16 | HBR; PfPR | gSG6-P1 IgG [%; L] |

*Table 1 continued on next page*

*Table 1 continued*

| Study year | Country | Malarial endemicity class | Dominant malaria vector species | Study design | No. participants (samples) | Study-specific n | Vector and malariometric variables | Salivary antibody outcomes (seroprevalence [%];[L]levels) |
|---|---|---|---|---|---|---|---|---|
| **Poinsignon et al., 2010b** | Senegal | Hypoendemic | An. funestus | Cohort‡ | 87 (261) | 3 | HBR; Plas+LM§, PfPR | gSG6-P1 IgG [L] |
| **Sarr et al., 2012** | Senegal | Hypoendemic; mesoendemic | An. gambiae s.l.; An. funestus† | Repeated cross-sectional‡ | - (401) | 4 | HBR; Plas+LM§, Pf-IgG; PfPR | gSG6-P1 IgG [%; L] |
| **Lawaly et al., 2012** | Senegal | Mesoendemic | An. gambiae s.l. | Cohort | 387 (711) | 4 | HBR; Plas+LM§, PfPR | gSGE IgG, IgG4 and IgE [L] |
| **Ali et al., 2012** | Senegal | Hypoendemic;* mesoendemic;* hyperendemic* | An. gambiae s.l.; An. funestus; An. pharoensis | Cross-sectional | - (134) | 3 | HBR; EIR | gSG6 IgG [%; L] fSG6 IgG [%; L]; f5'nuc IgG [%; L]; g5'nuc IgG [%; L] |
| **Ambrosino et al., 2010** | Senegal | Hypoendemic;* mesoendemic;* hyperendemic* | An. gambiae s.l.; An. funestus; An. pharoensis | Cross-sectional | - (123) | 3 | EIR; Pf-IgG | gSG6-P1 IgG [%]; gSG6-P2 IgG [%] |
| **Perraut et al., 2017** | Senegal | Hypoendemic; mesoendemic | An. gambiae s.l.; An. funestus | Repeated cross-sectional | - (798) | 4 | EIR; Plas+LM, Plas+PCR, Pf-IgG; PfPR | gSG6-P1 IgG [%] |
| **Poinsignon et al., 2008a** | Senegal | Mesoendemic | An. gambiae s.l. | Cross-sectional‡ | 241 (241) | 3 | HBR; PfPR | gSG6-P1 IgG [L]; gSG6-P2 IgG [L] |
| **Poinsignon et al., 2009** | Senegal | Mesoendemic | An. gambiae s.l. | Repeated cross-sectional‡ | 61 (122) | 2 | HBR; Plas+LM§, PfPR | gSG6-P1 IgG [L] |
| **Remoue et al., 2006** | Senegal | Mesoendemic | An. gambiae s.l. | Cross-sectional‡ | 448 (448) | 4 | HBR; Plas+LM§, PfPR | gSGE IgG [%; L] |
| **Sagna et al., 2019** | Senegal | Hypoendemic | An. gambiae s.l.† | Cross-sectional‡ | 809 (809) | 4 | PfPR | gSG6-P1 IgG [L] |
| **Stone et al., 2012** | Tanzania | Mesoendemic; hyperendemic | An. gambiae s.l. | Cross-sectional‡ | 636 (636) | 16 | HBR; Pf-IgG; PfPR | gSG6 IgG [%; L] |
| **Yman et al., 2016** | Tanzania | Mesoendemic; holoendemic* | An. gambiae s.l.; An. funestus | Repeated cross-sectional‡ | 668 (668) | 16 | Pf-IgG; PfPR | gSG6 IgG [%] |
| **Proietti et al., 2013** | Uganda | Mesoendemic | An. gambiae s.l.; An. funestus† | Repeated cross-sectional | 509 (509) | 3 | Pf-IgG; PfPR | gSG6 IgG [%] |
| *South America* | | | | | | | | |
| **Andrade et al., 2009** | Brazil | Eliminating; hypoendemic | An. darlingi | Cross-sectional | 204 (204) | 3 | Plas+LM¶, Plas+PCR¶, PfPR | dSGE IgG [L¶] |
| **Londono-Renteria et al., 2015a** | Colombia | | An. albimanus | Cross-sectional | 42 (42) | 2 | Plas+PCR¶ | gSG6-P1 IgG [L¶] |
| **Londono-Renteria et al., 2020a** | Colombia | Eliminating | An. albimanus | Cross-sectional | 337 (337) | 2 | Plas+PCR, PfPR | aPEROX-P1, P2 and P3 IgG [L]; aTRANS-P1 and P2 IgG [L] |
| **Montiel et al., 2020** | Colombia | Eliminating | An. albimanus | Case–control | 113 (113) | 2 | Plas+LM, Plas+PCR¶, PfPR | gSG6-P1 IgG [L¶]; dSGE IgG [L¶]; aSTECLA SGE IgG [L¶]; aCartagena SGE IgG [L¶] |
| *Asia* | | | | | | | | |
| **Kerkhof et al., 2016** | Cambodia | Hypoendemic | An. dirus | Cross-sectional | - (8438) | 113 | Plas+PCR, Pf-IgG, Pv-IgG; PfPR | gSG6-P1 IgG [%; L]; gSG6-P2 IgG [%; L] |

*Table 1 continued on next page*

*Table 1 continued*

| Study year | Country | Malarial endemicity class | Dominant malaria vector species | Study design | No. participants (samples) | Study-specific n | Vector and malariometric variables | Salivary antibody outcomes (seroprevalence [%];[L]levels) |
|---|---|---|---|---|---|---|---|---|
| **Charlwood et al., 2017** | Cambodia | Eliminating | An. dirus | Repeated cross-sectional | 454 (1180) | 6 | HBR; Plas+PCR, Pf-IgG; PfPR | gSG6 IgG [L] |
| **Ya-Umphan et al., 2017** | Myanmar | Eliminating | An. minimus; An. maculatus; An. dirus s.l. | Repeated cross-sectional | 2602 (9425) | 28 | HBR; EIR; Plas+PCR, Pf-IgG; PfPR | gSG6-P1 IgG [%; L] |
| **Waitayakul et al., 2006** | Thailand | Eliminating | An. dirus | Case–control | 139 (139) | 3 | Plas+LM | dirSGE IgG and IgM [L‖] |
| *Pacific* | | | | | | | | |
| **Pollard et al., 2019** | Solomon Islands | Eliminating; hypoendemic | An. farauti | Repeated cross-sectional | 686 (791) | 9 | HBR; EIR; PfPR | gSG6-P1 IgG [%; L] |
| **Idris et al., 2017** | Vanuatu | Eliminating; hypoendemic; mesoendemic | An. farauti | Repeated cross-sectional | 905 (905) | 3 | Plas+LM, Pf-IgG; Pv-IgG; PfPR | gSG6 IgG [%; L] |

Data are given as study, year of publication, country, malarial endemicity class, malarial DVS, study design (‡ indicates that study was performed solely in children), number of participants and number of samples, number of study-specific salivary antibody outcome observations (study-specific n), entomological and malariometric parameters, and salivary antibody outcomes assessed. Malarial endemicity class (categorical) is derived from *P. falciparum* prevalence rate in 2–10 year olds (PfPR) extracted from MAP using site geolocations and year of study, and applying established cut-offs reported in **Bhatt et al., 2015**. If PfPR data were not available (e.g. surveys prior to 2000; or unable to determine study site geolocation and year), endemicity class is given as stated in the study (indicated by *). DVS is as stated in the study or extracted from MAP (indicated by †). Of note, *An. gambiae* sensu lato (s.l.) includes both *An. gambiae* sensu stricto and *An. arabiensis*. Entomological and malariometric parameters include HBR, EIR, prevalence estimates of *Plasmodium* spp. (Plas+): detected by LM, or PCR, with § indicating prevalence of *P. falciparum* only and ¶ indicating prevalence of *P. vivax* only (no footnote indicates *P. falciparum* and *P. vivax* co-endemic), as well as PfPR extracted from MAP (**The Malaria Atlas, 2017**). Salivary antibody outcomes are indicated as either seroprevalence [%] or levels [L], or both [%; L], with ‖ indicating that studies reported results stratified by malarial infection status. Salivary antigens include recombinant full-length proteins, synthetic peptides, and whole SGE. Italicised prefix of salivary antigen indicates species: *An. gambiae* (g), *An. funestus* (f), *An. darlingi* (d), *An. albimanus* (a), *An. dirus* (dir).

DVS: dominant vector species; MAP: Malaria Atlas Project; HBR: human biting rate; EIR: entomological inoculation rate; LM: light microscopy; PCR: polymerase chain reaction; SGE: salivary gland extracts.

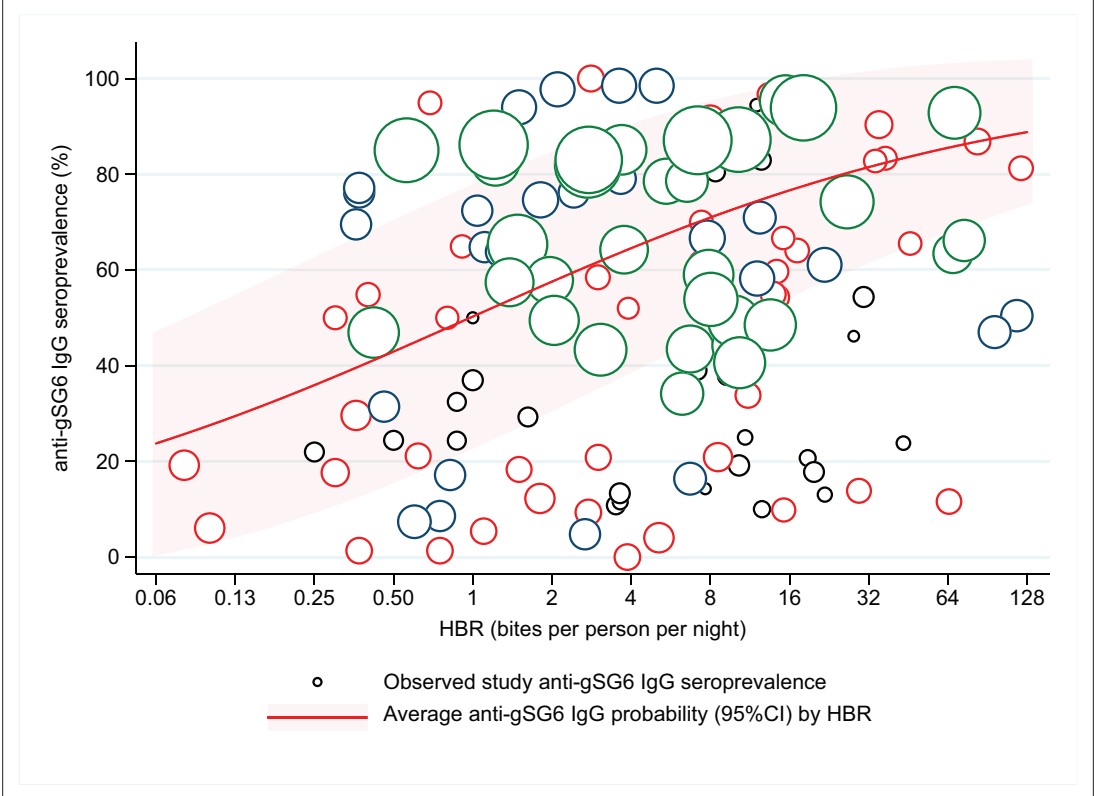

**Figure 2.** Association between anti-gSG6 IgG seroprevalence and log$_2$ human biting rate (HBR). Figure shows the observed anti-gSG6 (either recombinant or peptide form) IgG seroprevalence (%) and HBR for each study-specific observation, as well as the predicted average anti-gSG6 IgG seroprevalence (predicted probability for the average study and country) with 95% confidence intervals (95% CI). Circles are proportional to the size of the sample for each study-specific observation, with colours indicating sample size: black (<50), red (50–100), navy (100–150), and green (>150). Association estimated using generalised linear multilevel modelling (mixed effects, logistic) to account for the hierarchical nature of the data, where study-specific anti-gSG6 IgG observations are nested within study and study is nested within country (model output shown in Appendix 4; p<0.001).

The online version of this article includes the following figure supplement(s) for figure 2:

**Figure supplement 1.** Estimated relative change in odds of anti-gSG6 IgG seropositivity (95% confidence interval) for given relative percent increases in human biting rate (HBR) (bites/person/night).

odds of gSG6 IgG seropositivity compared to an attenuated association for African studies where *An. gambiae* s.l. was not the only DVS (OR: 1.14 per twofold increase in HBR; 95% CI: 0.98–1.33; p=0.079) and non-African studies where *An. gambiae* s.l. was absent (OR: 1.05 per twofold increase in HBR; 95% CI: 1.03–1.08; p<0.001). In order to quantify the relationship between gSG6 IgG seroprevalence and HBR, for given HBR values we estimated gSG6 IgG seroprevalence by producing model-based predicted probabilities overall and by DVS (*Figure 3*). In African studies where *An. gambiae* s.l. is the only DVS, predicted seroprevalence of *An. gambiae* gSG6 ranged from 21% (95% CI: 0–45%) to 86% (95% CI: 67–100%) for an HBR of 0.1–100 bites per person per night, respectively (*Figure 3*, *Figure 3—figure supplement 1*).

A positive association was also found between seroprevalence of anti-gSG6 IgG antibodies and EIR in analysis of n = 38 study-specific observations from eight studies (*Figure 4*, Appendix 6) [*Rizzo et al., 2011b*; *Ya-Umphan et al., 2017*; *Soma et al., 2018*; *Ali et al., 2012*; *Ambrosino et al., 2010*; *Perraut et al., 2017*; *Pollard et al., 2019*; *Badu et al., 2012b*]. For a twofold increase in EIR, the odds of anti-gSG6 IgG seropositivity increased by 11% (OR: 1.11; 95% CI: 1.05–1.17; p<0.001), with heterogeneity in the study-specific effects (95% reference range: 1.00–1.24; likelihood ratio $\chi^2(1)$ = 15.02, p<0.001). There was no evidence of effect modification by either vector collection method (p=0.095) or DVS (p=0.080) on the association between seroprevalence of anti-gSG6 IgG and EIR.

Similar positive associations were also found between anti-gSG6 IgG levels, HBR, and EIR in 11 studies [*Drame et al., 2015*; *Drame et al., 2012*; *Stone et al., 2012*; *Soma et al., 2018*; *Ali et al.,*

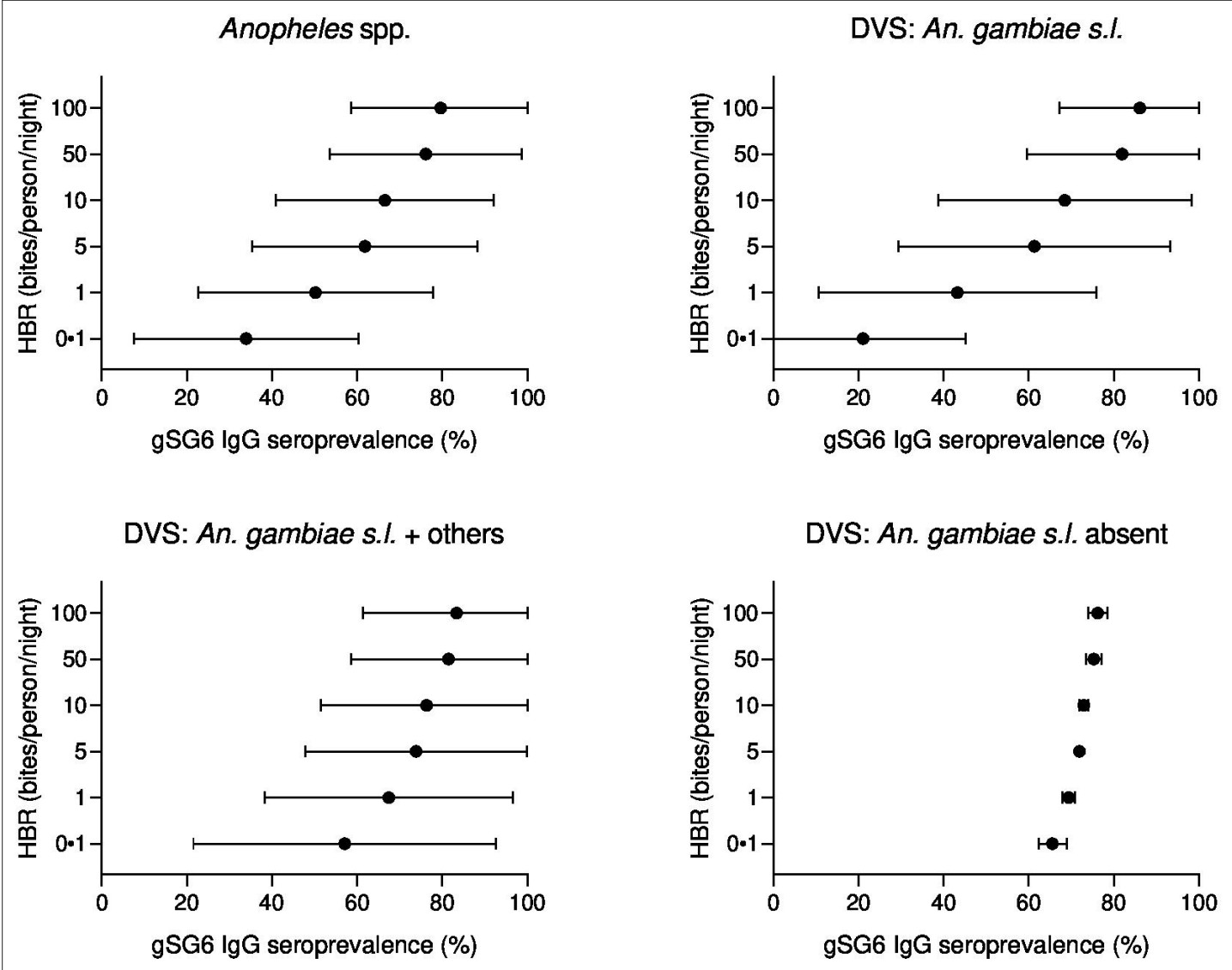

**Figure 3.** Forest plots of predicted anti-gSG6 IgG seroprevalence (%) and *Anopheles* species-specific human biting rate (HBR). Panels show the predicted average anti-gSG6 IgG seroprevalence (predicted probability for the average study and country) with 95% confidence intervals for given HBR, for all *Anopheles* spp. (using model output from Appendix 4) and for specific-dominant vector species (DVS): where *An. gambiae* s.l. is the only DVS, where other DVS were present in addition to *An. gambiae* s.l. and where *An. gambiae* s.l. was absent (using model output from Appendix 5).

The online version of this article includes the following figure supplement(s) for figure 3:

**Figure supplement 1.** Association between anti-gSG6 IgG seroprevalence and *Anopheles* species-specific $log_2$ human biting rate (HBR).

---

*2012*; *Rizzo et al., 2011a*; *Rizzo et al., 2011b*; *Poinsignon et al., 2010b*; *Poinsignon et al., 2008a*; *Charlwood et al., 2017*; *Sagna et al., 2013b*] and 3 studies [*Rizzo et al., 2011b*; *Ya-Umphan et al., 2017*; *Ali et al., 2012*], respectively, but 7 studies showed no association between HBR and levels of IgG to gSG6 [*Drame et al., 2010a*; *Ya-Umphan et al., 2017*; *Traoré et al., 2018*; *Traoré et al., 2019*; *Sarr et al., 2012*; *Poinsignon et al., 2009*; *Pollard et al., 2019*].

The association between anti-gSG6 IgG seroprevalence and population-level prevalence of *Plasmodium* spp. infection was investigated. Generalised linear multilevel modelling (mixed effects, logistic) of n = 212 from 14 studies that measured *Plasmodium* spp. prevalence contemporaneously in their study [*Badu et al., 2012b*; *Rizzo et al., 2011b*; *Ya-Umphan et al., 2017*; *Soma et al., 2018*; *Koffi et al., 2015*; *Traoré et al., 2019*; *Badu et al., 2015*; *Sagna et al., 2013b*; *Sarr et al., 2012*; *Perraut et al., 2017*; *Drame et al., 2010a*; *Idris et al., 2017*; *Proietti et al., 2013*; *Kerkhof et al., 2016*] showed that for a twofold increase in the prevalence of *Plasmodium* spp. infection the odds

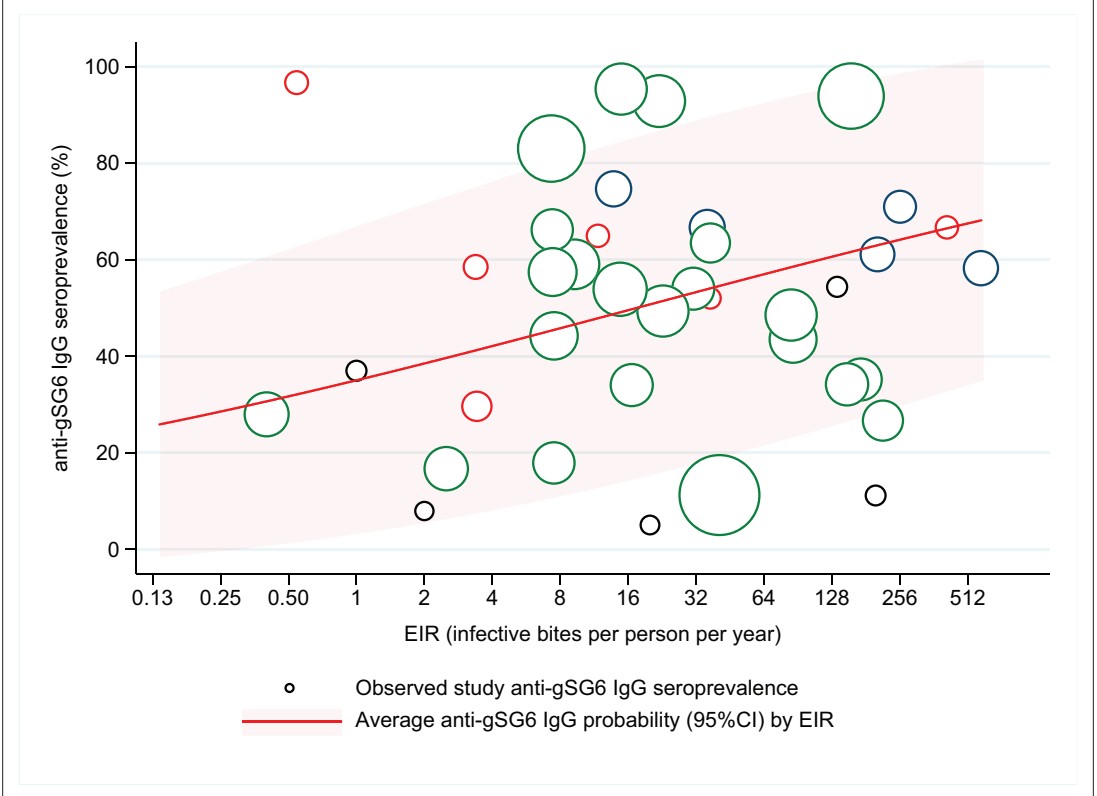

**Figure 4.** Association between anti-gSG6 IgG seroprevalence and log$_2$ entomological inoculation rate (EIR). Figure shows the observed anti-gSG6 (either recombinant or peptide form) IgG seroprevalence (%) and EIR for each study-specific observation, as well as the predicted average anti-gSG6 IgG seroprevalence (predicted probability for the average study and country) with 95% confidence intervals (95% CI). Circles are proportional to the size of the sample for each study-specific estimate, with colours indicating sample size: black (<50), red (50–100), navy (100–150), and green (>150). Association estimated using generalised linear multilevel modelling (mixed effects, logistic) to account for the hierarchical nature of the data, where study-specific anti-gSG6 IgG observations are nested within study and study is nested within country (model output shown in Appendix 6; p<0.001).

The online version of this article includes the following figure supplement(s) for figure 4:

**Figure supplement 1.** Estimated change in odds of anti-gSG6 IgG seropositivity (95% confidence interval) for given relative percent increases in entomological inoculation rate (EIR) (infective bites/person/night).

of gSG6 IgG seropositivity increased by 38%, although the confidence intervals were wide (OR: 1.38; 95% CI: 0.89–2.12; p=0.148) and heterogeneity in the study-specific effects was observed (95% reference range: 0.30–6.37; likelihood ratio $\chi^2(1)$ = 235.5, p<0.001) (*Figure 5* and Appendix 7). In the association between gSG6 IgG seropositivity and *Plasmodium* spp. infection, there was no evidence for a moderating effect of *Plasmodium* spp. detection method (light microscopy or PCR, p=0.968), or species (African studies with *P. falciparum* versus non-African studies where *P. falciparum* and *P. vivax* are co-prevalent, p=0.538).

Additionally, 14 studies reported observations of anti-gSG6 IgG levels and the prevalence of *Plasmodium* spp. infections measured contemporaneously in their study. The median anti-gSG6 IgG antibody levels increased with increasing *Plasmodium* spp. prevalence in six of these studies (*Drame et al., 2010a*; *Ya-Umphan et al., 2017*; *Idris et al., 2017*; *Poinsignon et al., 2010b*; *Sarr et al., 2012*; *Kerkhof et al., 2016*), or in *Plasmodium* spp.-infected compared to non-infected individuals (*Londono-Renteria et al., 2015a*; *Montiel et al., 2020*), but showed no association in eight studies (*Rizzo et al., 2011b*; *Soma et al., 2018*; *Koffi et al., 2015*; *Traoré et al., 2018*; *Traoré et al., 2019*; *Badu et al., 2015*; *Sagna et al., 2013b*; *Poinsignon et al., 2009*). Furthermore, we also investigated associations with serological measures of malaria exposure and found that for a twofold increase in pre-erythrocytic and blood stage antigen seroprevalence there was a 2.19-fold (OR: 2.19; 95% CI: 1.18–4.04; p=0.013) and 41% to 5.69-fold (OR range: 1.41–5.69; p range: <0.001 to 0.523) increase in the odds of anti-gSG6 IgG seropositivity, respectively (Appendix 8).

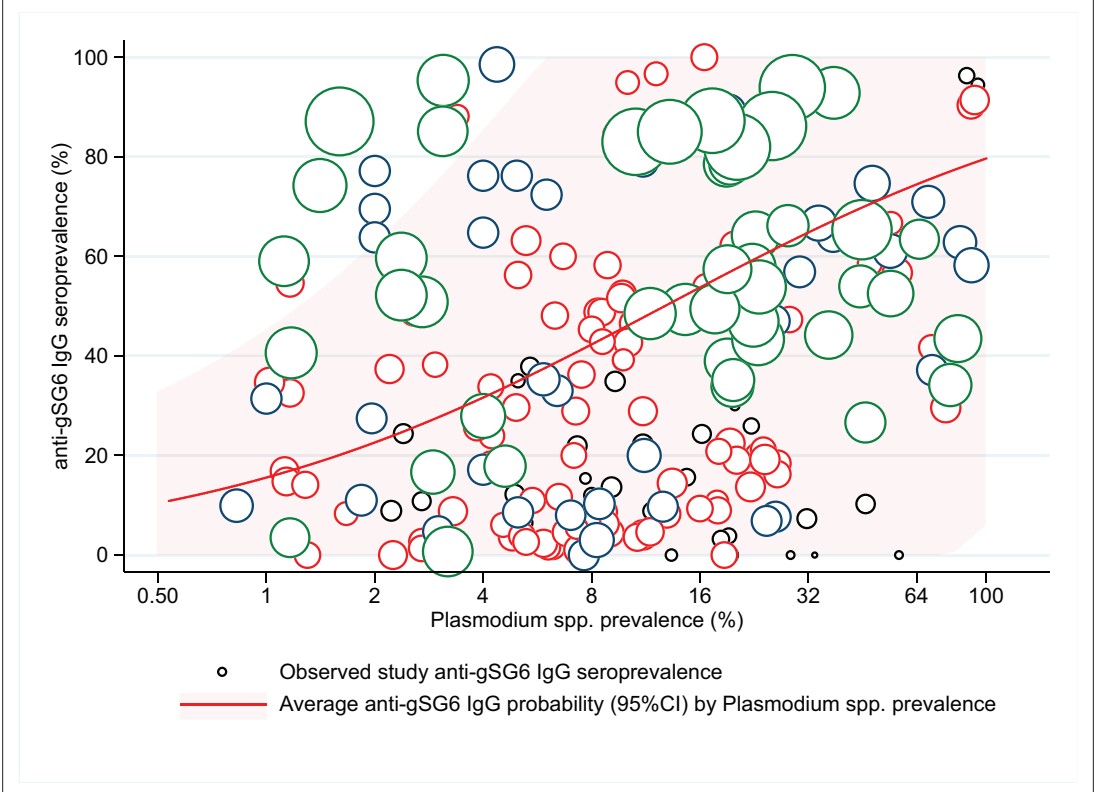

**Figure 5.** The association between anti-gSG6 IgG seroprevalence (%) and log$_2$ *Plasmodium* spp. prevalence (%). Figure shows the observed anti-gSG6 (either recombinant or peptide form) IgG seroprevalence (%) and prevalence of any *Plasmodium* spp. infection (%) for each study-specific observation, as well as the predicted average anti-gSG6 IgG seroprevalence (predicted probability for average study) with 95% confidence intervals (95% CI). Circles are proportional to the size of the sample for each study-specific observation, with colours indicating sample size: black (<50), red (50–100), navy (100–150), and green (>150). Association estimated using generalised linear multilevel modelling (mixed effects, logistic) to account for the hierarchical nature of the data, where study-specific anti-gSG6 IgG observations are nested within study. See Appendix 7 for model output.

To give epidemiological context, we estimated anti-gSG6 seroprevalence by producing model-based predicted probabilities by malarial endemicity class (a categorical variable derived by applying established cut-off values for the *Pf*PR$_{2-10}$ extracted from MAP). Generalised linear multilevel modelling (mixed effects, logistic) on 297 study-specific salivary antibody observations from 22 studies shows that the estimated anti-gSG6 IgG seroprevalence is higher for the higher endemicity classes (eliminating malaria: 20% [95% CI: 8–31%]; hypoendemic: 34% [95% CI: 19–49%]; mesoendemic: 52% [95 CI: 35–68%]; hyperendemic settings: 47% [95% CI: 27–64%]; holoendemic: 78% [95% CI: 67–90%]; p<0.001; *Table 2*). Interactions with DVS or region (Africa/non-Africa) could not be explored due to collinearity with malaria endemicity class. Therefore, in addition using Bayes best linear unbiased predictions (BLUPs) we estimated country-specific gSG6 IgG seroprevalence from an intercept-only multilevel model fitted to 301 study-specific salivary antibody observations from 22 studies. It showed that IgG seroprevalence to *An. gambiae* gSG6 was lowest in countries in the Pacific region where *An. gambiae* is absent (Vanuatu [31%] and Solomon Islands [32%]) and highest in countries where *An. gambiae* is a DVS (Benin [72%] and Burkina Faso [65%]; Appendix 9).

Assessments of internal and external study validity revealed there was a moderate risk of selection bias (Appendix 2) due to the study-specific inclusion criteria of populations at higher risk of malaria which contributed gSG6 seroprevalence observations. Sensitivity analyses exploring potential study-level outlier influence on the estimated associations between anti-gSG6 IgG seroprevalence, HBR, and EIR showed no evidence of bias (effect estimates for each sensitivity analysis were consistent with model estimates overall) for studies identified as exhibiting potential influence (HBR: n = 6; EIR: n = 6).

**Table 2.** Association between gSG6 IgG seroprevalence (%) and malarial endemicity ($PfPR_{2-10}$). Table shows the odds ratio (OR), 95% confidence interval (95% CI), p-value, as well as the predicted gSG6 IgG seroprevalence and associated 95%CI[†] for associations between endemicity class (categorical: derived from *P. falciparum* parasite rates in 2–10 year olds [*Pf*PR]) and anti-gSG6 IgG seropositivity.

| Malaria endemicity class* | OR | 95% CI | p-Value | Predicted gSG6 IgG seroprevalence (%) | 95% CI |
|---|---|---|---|---|---|
| Eliminating malaria(*PfPR <1%*) | Ref. | | | 20.0 | 8.3–31.7 |
| Hypoendemic(*PfPR 1-10%*) | 2.04 | 1.43–2.90 | <0.001 | 33.7 | 18.9–48.5 |
| Mesoendemic(*PfPR 10-50%*) | 4.19 | 2.80–6.08 | <0.001 | 51.5 | 34.6–67.7 |
| Hyperendemic(*PfPR 50-75%*) | 3.36 | 1.98–5.71 | <0.001 | 46.5 | 27.4–63.8 |
| Holoendemic(*PfPR >75%*) | 14.4 | 9.72–21.36 | <0.001 | 78.2 | 66.8–89.7 |

*Generalised linear multilevel modelling (mixed effects, logistic) estimating the association between anti-gSG6 IgG seropositivity and endemicity class with random effects for study-specific heterogeneity in gSG6 IgG. Model fitted to n = 297 study-specific observations from 22 studies. Of note, nine studies that measured *Plasmodium* spp. prevalence and IgG antibodies to gSG6 were excluded from this analysis as eight only reported gSG6 IgG levels and one was a case–control study. Endemicity class membership is derived from *Pf*PR from **The Malaria Atlas, 2017** (MAP) using cut-offs taken from **Bhatt et al., 2015**, or where MAP data were unavailable, endemicity was included as indicated in the study.

[†]Predicted gSG6 IgG seroprevalence (predicted probability in the average study) is estimated from generalised linear multilevel modelling (mixed effects, logistic).

## Discussion

This systematic review and multilevel modelling analysis provides the first quantification of a positive non-linear association between seroprevalence of *An. gambiae* gSG6 IgG antibodies and HBR and demonstrated that its magnitude varied with respect to the DVS present in the area. Importantly, this review identified a paucity of studies conducted outside of Africa, as well as investigating salivary antigens representing different *Anopheles* spp. and antigenic targets. gSG6 antibodies were positively associated with the prevalence of *Plasmodium* spp. infection as well as established epidemiological measures of malaria transmission: malaria endemicity class and EIR. Overall, our results demonstrate that antibody seroprevalence specific for *Anopheles* spp. salivary antigens has the potential to be an effective measure of vector exposure and malaria transmission at the population and, potentially, individual level.

*An. gambiae* gSG6 IgG seropositivity increased with increasing HBR, although these increases had diminishing impact on *An. gambiae* gSG6 IgG seropositivity at higher levels of HBR (approximately greater than two bites per person per night). In our study, 17 studies performed across Africa (Angola, Benin, Burkina Faso, Cote d'Ivoire, and Senegal) and the Asia Pacific (Cambodia, Myanmar, and the Solomon Islands) reported an HBR < 2, demonstrating the applicability of gSG6 as a biomarker of HBR across a broad range of malaria-endemic regions. We also observed that the association was strongest in areas where *An. gambiae* s.l. was the only DVS (i.e. concordant *An. gambiae* species-specific HBR with *An. gambiae* gSG6 antibodies). Associations, albeit weaker, were also observed between discordant species-specific HBR and gSG6, most likely because the *An. gambiae SG6* gene shares moderate sequence identity with vector species that are dominant in other regions (Africa: 80% *An. funestus*; Asia: 79% *Anopheles stephensi* and *Anopheles maculatus*; 54% *An. dirus*; Pacific: 52.5% *Anopheles farauti*), and is absent from the DVS of the Americas (*An. albimanus* and *An. darlingi*) (**Arcà et al., 2017**). The generalisability of *An. gambiae* gSG6 IgG as a biomarker of exposure to other *Anopheles* spp. may therefore be limited. However, our review also identified a paucity of studies investigating additional salivary antigenic targets and *Anopheles* species not present in Africa. The identification of novel salivary antigens that are species-specific will be valuable in quantifying exposure to the other *Anopheles* vectors that share limited identity with *An. gambiae SG6* (such as *An. farauti* and *An. dirus*), as well as *Anopheles* spp. which lack *SG6* (as done for *An. albimanus* and *An. darlingi*; **Londono-Renteria et al., 2020a**; **Londono-Renteria et al., 2020b**). An *Anopheles* species-specific serological platform could advance vector surveillance by more accurately capturing exposure to DVS in the South American and Asia Pacific regions which exhibit diverse biting behaviours and vector

competence (DVS typically bite outdoors during the night and day, respectively; *The Malaria Atlas, 2017*; *Sinka et al., 2012*; *Sinka et al., 2010*; *Trung et al., 2005*; *Herrera et al., 2015*; *Chaumeau et al., 2018*), as well as the increasing threat of urban malaria from *An. stephensi* in Africa (*Takken and Lindsay, 2019*; *Sinka et al., 2020*).

This review demonstrated that the prevalence of *Anopheles* salivary antibodies increased with increasing prevalence of *Plasmodium* spp. infection (although confidence intervals were wide and we observed heterogeneity in the effect between studies) as well as established epidemiological measures of malaria transmission: malaria endemicity class and EIR. Anti-salivary antibodies, such as SG6 IgG, may therefore have the potential to serve as a proxy measure for receptivity of a population to sustain malaria transmission. Their application could be particularly relevant in pre-elimination areas, or non-endemic areas under threat of imported malaria, where *Anopheles* salivary antibodies are more readily detectable than parasites; salivary antibodies were predicted to be prevalent (20%) in areas defined as eliminating malaria (<1% $Pf$PR$_{2\text{-}10}$). Furthermore, if SG6 IgG seroprevalence can be effectively combined with a measurement of the sporozoite index, salivary antibodies as a marker of HBR could help overcome sensitivity limitations of EIR in low transmission areas. Additional measures could include estimates of malaria prevalence or serological biomarkers that are species- or life stage-specific (e.g. *Plasmodium* spp. pre-erythrocytic antigens as biomarkers for recent parasite inoculation). Indeed, positive associations between antibodies specific for *Plasmodium* spp. pre-erythrocytic and blood stage antigens with gSG6 were demonstrated in analyses of data from diverse malaria-endemic areas. Serological tools combining salivary antigens with antigens specific for the different *Plasmodium* spp. could be easy to employ and complement malaria surveillance programmes. These tools may be particularly useful in the Asia Pacific, a region of relatively low malaria transmission with goals of elimination, but the highest burden of *P. vivax* malaria where blood stage infection can be caused by relapses from dormant liver stages. In these areas, parasite prevalence may therefore overestimate ongoing malaria transmission, making vector surveillance tools essential to informing elimination strategies in the Asia Pacific and other regions where *P. vivax* is endemic.

The gold standard entomological measures HBR and EIR provide crude population-level estimates of vector and malaria exposure that are specific in space and time and preclude investigation of individual-level heterogeneity and natural transmission dynamics. Our study demonstrated that salivary biomarkers measured at the individual level, such as gSG6 IgG, can be used to quantify total vector exposure at the population level, without requiring laborious entomological experiments. However, validating an individual-level serological measure, which demonstrates considerable individual-level variation, against the imperfect population-level gold standards of HBR and EIR is challenging and reflected in the variation in study-specific estimates in the association between gSG6 IgG and HBR in modelling analyses. However, the accuracy of salivary antibodies to measure individual-level exposure to *Anopheles* bites is yet to be validated; literature searches identified no studies investigating this association at the individual level. Without detailed measurements of individual-level vector exposure, or a detailed knowledge of the half-life of *Anopheles* salivary antibodies post biting event, the true accuracy of salivary antibodies, such as SG6 IgG, to measure individual-level HBR remains unknown. This knowledge is particularly pertinent where *Anopheles* salivary biomarkers might be applied to assess the effectiveness of a vector control intervention or used to measure temporal changes in malaria transmission; particularly in areas or populations where there is considerable heterogeneity in individual-level risk of *Anopheles* exposure (e.g. unmeasured outdoor biting due to occupational exposure for forest workers; *Sandfort et al., 2020*).

The broad nature of our inclusion and quality criteria was a key strength of our systematic review, which aimed to provide a comprehensive analysis of all *Anopheles* salivary biomarkers and determine their associations with entomological and malariometric measures of transmission. However, this review has two main limitations. First, despite the inclusive nature, assessment of the external validity of the review revealed a moderate risk of bias; some studies exhibited a high risk of selection bias as they were performed in specific high-risk populations not representative of the overall population (i.e. children only). This is accounted for to some degree by specification of a random effect (i.e. intercept) for study, which accounts for unmeasured study-specific factors that may introduce study-specific measurement error to measurement of the outcome. Second, with respect to internal validity, there may be potential selection bias introduced by the exclusion of studies reporting zero HBR (7 observations from three studies; *Rizzo et al., 2011b*; *Pollard et al., 2019*; *Sagna et al.,*

*2013b*), EIR (22 observations from three studies; *Ya-Umphan et al., 2017*; *Rizzo et al., 2011b*; *Soma et al., 2018*), and malaria prevalence (15 observations from three studies; *Idris et al., 2017*; *Sagna et al., 2013b*; *Kerkhof et al., 2016*) estimates, given we modelled the log of these factors. However, adding a small constant (e.g. 0.001) to a zero value to permit modelling of a log estimate can also introduce considerable bias (i.e. seemingly small differences between values become very large on the log scale). In light of this, we also chose to provide estimates of association and gSG6 IgG sero-prevalence according to a selected range of epidemiologically relevant hypothetical HBRs (no widely accepted HBR classification exists in the literature) and according to widely accepted, discrete, ende-micity classes according to MAP estimates (which permitted inclusion of all studies) to provide epide-miological context. However, there is the potential for misclassification of malarial endemicity class derived from geospatially extracted MAP predictions of $PfPR_{2-10}$ which increase in uncertainty in areas with scarce data. Similarly, we used MAP vector occurrence data to inform DVS categories for 7 (out of 42) studies. Cross-referencing these 7 studies with a 2017 updated database for African vectors (using data for the nearest neighbouring village) identified 10 discrepant datapoints from 3 studies (from a total of 28 datapoints from 7 studies) (*Snow, 2017*). Any misclassification events may cause us to underestimate the standard error in the effect of malaria endemicity class and DVS on gSG6 IgG.

## Conclusions

In order to advance progress towards malaria elimination, the World Health Organization has called for innovative tools and improved approaches to enhance vector surveillance and monitoring and evaluation of interventions (*World Health Organization, 2017*). Our systematic review has provided evidence that *Anopheles* salivary antibodies are serological biomarkers of vector and malaria expo-sure, by quantifying their positive association with *Anopheles*-HBR and established epidemiological measures of malaria transmission. These salivary biomarkers have the potential to replace crude population-level estimates of entomological indices with a precise and scalable tool that measures *Anopheles* vector exposure at the individual level. This approach could be expanded into a sero-surveillance tool to assess the effectiveness of vector control interventions, define heterogeneity in malaria transmission, and inform efficient resource allocation that would ultimately accelerate prog-ress towards elimination.

## Acknowledgements

We thank Franck Remoue, Anne Poinsignon, Bruno Arcà, Vincent Corbel, Richard Paul, André Sagna, Kingsley Badu, Berlin Londono-Renteria, Jacques Derek Charlwood, William Stone, Chris Drakeley, Karen Kerkhof, Sylvie Manguin, and Yunita Armiyanti for responding to requests for further informa-tion/data for the systematic review.

## Additional information

### Funding

| Funder | Grant reference number | Author |
| --- | --- | --- |
| National Health and Medical Research Council | 1134989 | Julie A Simpson Freya JI Fowkes |
| National Health and Medical Research Council | 1166753 | Freya JI Fowkes |
| National Health and Medical Research Council | 1196068 | Julie A Simpson |
| Australian Government | Australian Government Research Training Program Scholarship | Ellen A Kearney |
| Wellcome Trust | 220211 | Victor Chaumeau |

| Funder | Grant reference number | Author |
|---|---|---|
| Victorian State Government | Operational Infrastructure Support Program received by the Burnet Institute. | Ellen A Kearney Paul A Agius Julia C Cutts Freya JI Fowkes |

The funders had no role in study design, data collection and interpretation, or the decision to submit the work for publication.

## Author contributions

Ellen A Kearney, Data curation, Formal analysis, Investigation, Methodology, Project administration, Visualization, Writing – original draft, Writing – review and editing; Paul A Agius, Formal analysis, Investigation, Methodology, Supervision, Visualization, Writing – original draft, Writing – review and editing; Victor Chaumeau, Julia C Cutts, Julie A Simpson, Methodology, Writing – review and editing; Freya JI Fowkes, Conceptualization, Data curation, Formal analysis, Funding acquisition, Investigation, Methodology, Project administration, Supervision, Writing – original draft, Writing – review and editing

## Author ORCIDs

Ellen A Kearney http://orcid.org/0000-0001-8912-8067
Victor Chaumeau http://orcid.org/0000-0003-0171-2176
Julie A Simpson http://orcid.org/0000-0002-2660-2013
Freya JI Fowkes http://orcid.org/0000-0001-5832-9464

## Decision letter and Author response

Decision letter https://doi.org/10.7554/eLife.73080.sa1
Author response https://doi.org/10.7554/eLife.73080.sa2

---

# Additional files

## Supplementary files

• Transparent reporting form

## Data availability

The current manuscript is a systematic review with multilevel modelling of study level data. The constructed dataset and associated code used for analyses are available at https://github.com/ellenakearney/Anopheles_salivary_biomarker_systematic_review (copy archived at swh:1:rev:ead-b7ab1cbf93c730fb463f43e1b13b9ae16ddd3). Data for 'Dominant malaria vector species globally, 2010' (Sinka et al., 2012) and 'Plasmodium falciparum parasite rate in 2-10 year olds globally, 2000-2017' (Weiss et al., 2019) can be accessed from the Malaria Atlas Project Explorer (https://malariaatlas.org/explorer/#/).

The following dataset was generated:

| Author(s) | Year | Dataset title | Dataset URL | Database and Identifier |
|---|---|---|---|---|
| Kearney EA, Agius PA, Chaumeau V, Cutts JC, Simpson JA, Fowkes FJI | 2021 | Anopheles salivary biomarkers review.dta | https://github.com/ellenakearney/Anopheles_salivary_biomarker_systematic_review | GitHub, Anopheles_salivary_biomarker_systematic_review |

---

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

# Appendix 1

## Supplementary methodology

### Search strategy

We performed a systematic review with multilevel modelling of the published literature according to the MOOSE guidelines (*Stroup et al., 2000*) and the PRISMA specifications (*Moher et al., 2009*). The protocol was registered with PROSPERO (CRD42020185449).

The electronic databases PubMed, Scopus, Web of Science, African Index Medicus, and the Latin American and Caribbean Health Sciences Literature (LILACS) were searched for studies published before 30 June 2020 investigating *Anopheles* salivary antigens as a biomarker for mosquito exposure or malaria transmission. Search terms were as follows: *Anophel* AND saliva* AND (antibod* OR sero* OR antigen OR marker* OR biomarker* OR gSG6* OR gSG* OR SG* OR cE5). The reference lists of included studies were screened for additional studies, and Google Scholar was used to identify additional works by key authors. No formal attempt was made to identify unpublished population studies as it would have required significant description of the design, methods, and analysis used in these studies, and a review of ethical issues.

### Selection criteria

The primary criterion for inclusion in this systematic review was the reporting of observations of seroprevalence or total levels of Ig antibodies (including all isotypes and subclasses) in human sera against recombinant or synthetic peptide *Anopheles* salivary antigens. We considered for inclusion cross-sectional studies, cohort studies, intervention studies, and case–control studies of individuals or populations (including sub-populations) living in all geographies with natural exposure to *Anopheles* mosquitoes. Studies that were solely performed in participants not representative of the wider population (i.e. mosquito-allergic patients, soldiers, returned travellers) were excluded. The minimum quality criteria for inclusion in this review were antibody detection performed using enzyme-linked immunosorbent assay (ELISA), multiplex or Luminex assays.

The exposure variables of interest included entomological and malariometric parameters, including (i) HBR, defined as the number of bites received per person per unit of time; (ii) EIR, defined as the number of infectious bites per person per unit of time, calculated as the HBR multiplied by the sporozoite index; (iii) estimates of malaria prevalence; and (iv) population-level seroprevalence estimates against *Plasmodium* spp. malarial antigens. To ensure HBR estimates were given for the same unit of time (bites per person per night), biting rates given per week were divided by 7, and biting rates given per month we multiplied by 12 and divided by 365. Similar approaches were employed to ensure consistent units for EIR (infectious bites per person per year). *Plasmodium* spp. infections had to be confirmed by either microscopy, RDT, or molecular methods (PCR). *Plasmodium* spp. diagnosis was included for all *Plasmodium* spp. combined and the species level if provided. Where exposure estimates were not provided, we attempted to source data from other publications by the authors or used the site geolocation and year to obtain estimates of EIR from the Pangaea dataset (*Yamba et al., 2018*). *P. falciparum* rates in 2–10 year olds (globally, 2000–2017) and DVS from the MAP (*The Malaria Atlas, 2017*). Studies of salivary antigens where exposure variables could not be sourced and data that could not be extracted were excluded.

### Selection of studies

One author performed database searches and screened reference lists to identify possible studies. One author screened studies against inclusion criteria, with discussion and input from a second reviewer.

### Approaches to include all available studies

The authors of any studies that did not contain relevant information on the study design, populations, eligibility criteria, or key study data were contacted and relevant data requested. Authors were contacted via an initial email detailing the precise nature of the systematic review and the data required. If the authors did not reply to three email requests or were unable to provide relevant data, the studies were deemed to insufficiently meet inclusion/quality criteria and were excluded. As measurement of antibody levels does not produce a common metric between studies, authors were asked to classify their participants as 'responders' or 'no-responders' according to seropositivity

(antibody level relative to unexposed sera) within each study to allow comparisons of seroprevalence between studies (*Cutts et al., 2020*; *Cutts et al., 2014*; *Fowkes et al., 2010*). Studies that were only able to provide antibody levels or categorised seropositivity based upon arbitrary cut-offs were excluded from multilevel modelling analyses and included in narrative terms. Where the salivary antibody response and exposure variable were measured in the same population and reported in multiple publications, the study with the largest sample size was included, otherwise the earliest study was included.

## Data extraction

Data were extracted using a data collection form by one reviewer. Any data that was provided at the sub-population level was extracted at the lowest level, that is, if a study was performed across multiple sites, and an estimate for both salivary antibody seroprevalence/levels and the exposure of interest is given for each site, it was included the site level, rather than an aggregated level.

## Measures

### Outcomes

The primary outcome of interest of our systematic review was the reported antibody response (both seroprevalence and levels of all Ig subclasses and isotypes) to *Anopheles* salivary antigens. Multilevel modelling analyses were performed where the seroprevalence of antibodies against the same antigen and the exposure of interest were reported in more than one study.

### Exposures

The primary exposures of interest included in the multilevel modelling analyses were the HBR and EIR, a measure of the average number of bites received per person per night and infectious bites received per person per year, respectively. Secondary exposures assessed include the prevalence of any *Plasmodium* spp. infection (including *P. falciparum* only, *P. vivax* only, or untyped infections). Additional secondary exposures include the *P. falciparum* infection rate in 2–10 year olds extracted from MAP, as well as the seroprevalence of antimalarial antibodies against pre-erythrocytic and blood stage antigens.

Clinical and methodological heterogeneity were explored using prespecified variables to minimise spurious findings. Variables considered for inclusion were study design (cohort, cross-sectional, repeated cross-sectional), DVS, study participants (adults only, children only, adults and children), preparation of salivary antigen (recombinant full-length protein, synthetic peptide), malaria detection methodology (light microscopy, RDT, PCR), and entomological vector collection methodology (human landing catch, light traps, and spray catches).

### Statistical analysis

Where there were sufficient data to pool observations of the same exposure and outcome measures, generalised linear multilevel modelling was used to undertake analyses quantifying associations between the exposures of interest and salivary antibody seroprevalence measurements. Models were generalised through use of the logit link function and binomial distribution (statistical notation for HBR model shown below as Equation 1). Seroprevalence was modelled in binomial form as the number of individuals seropositive to the total sample size. A three-level random effects model with a nested framework was used to account for dependency in the data, with random intercepts for country (level 3) and study (level 2) estimated. Hence, level 1 units represented multiple salivary antibody observations within a study induced by the study design (i.e. multiple time points, sites, age categories). Additionally, study-level random slopes for entomological and malariometric exposures were estimated to permit the effects to vary across studies. Model structure was determined empirically through likelihood ratio tests (p<0.05), with the exception of country at the third,level which was included in HBR and EIR analyses to estimate country-specific seroprevalence estimates of anti-salivary antibodies. The associations between the various exposures and the different salivary antigens were analysed separately; however, observations of IgG seroprevalence against the recombinant full-length protein (gSG6) and synthetic peptide (gSG6-P1, the one peptide determined in all studies utilising peptides) form of the gSG6 antigen were analysed together, with a fixed term for antigen construct considered for inclusion in the model. Of note, gSG6 peptide 2 (gSG6-P2) was excluded from being analysed with gSG6 and gSG6-P1 as the two studies that reported anti-gSG6-P2 IgG seroprevalence also reported the seroprevalence of anti-gSG6-P1 IgG and only one could be included. Potential effect modification of the associations between the exposures of interest and the anti-*Anopheles* salivary antibody responses was explored and undertaken by estimating interaction terms for DVS (*An. gambiae* s.l. only, *An. gambiae* s.l. and other DVS, or *An. gambiae* s.l. absent) and for vector collection method (human landing catch or other indirect measures, e.g. light traps, spray catches, etc.). For the association between *Plasmodium* spp. prevalence and gSG6 IgG seropositivity, interaction terms for malaria detection methodology (light microscopy or PCR), and malarial species type (*P. falciparum* only, or *P. falciparum* and *P. vivax*) were estimated. Other variables considered for inclusion in adjusted models were study design, participant, and salivary antigen construct; however, these variables showed no association with anti-gSG6 IgG and were thus excluded.

AIC and BIC fit indices were used to determine the best-fitting functional forms for the association between log odds of gSG6 IgG seropositivity and HBR, EIR, and *Plasmodium* spp. prevalence – linear, log, quadratic, and cubic functions were fitted, with a log transformation exhibiting superior model fit (*Appendix 1—table 1*). To aid interpretation, we present our results as a relative increase

in the odds of the gSG6 IgG seropositivity for a twofold (100% relative) increase in the exposures. Additional relative percent changes in HBR and EIR are also presented.

**Appendix 1—table 1.** Model selection process, showing the log likelihood, Akaike's information criterion (AIC), and Bayesian information criterion (BIC) fit indices for each model estimating different functional forms for the association between gSG6 IgG seropositivity and respective exposures.

| Model | Log likelihood | AIC | BIC |
|---|---|---|---|
| *HBR* | | | |
| Linear | −1533.3 | 3076.6 | 3091.2 |
| *Log* | **−1492.8** | **2995.7** | **3010.1** |
| Quadratic | −1523.7 | 3059.4 | 3077.0 |
| Cubic | −1523.7 | 3061.3 | 3081.9 |
| | | | |
| *EIR* | | | |
| Linear | −1003.40 | 2016.80 | 2027.27 |
| *Log* | **−530.65** | **1071.30** | **1079.49** |
| Quadratic | −1002.65 | 2017.30 | 2029.87 |
| Cubic | −976.36 | 1966.72 | 1981.38 |
| | | | |
| *Plasmodium* spp. prevalence | | | |
| Linear | −2777.45 | 5564.91 | 5582.03 |
| *Log* | **−2597.24** | **5202.47** | **5215.90** |
| Quadratic | −2775.47 | 5562.95 | 5583.50 |
| Cubic | −2769.91 | 5553.82 | 5577.80 |

HBR: human biting rate; EIR: entomological inoculation rate.

Empirical Bayes BLUPs were used to estimate the probability of gSG6 IgG seropositivity in the average study and country, which is equivalent to an estimated gSG6 IgG seroprevalence. In order to maximise the number of included studies in our modelling, we predicted anti-gSG6 seroprevalence according to endemicity class, derived by applying established endemicity cut-offs to $PfPR_{2\text{-}10}$ estimates (*Bhatt et al., 2015*) extracted from MAP using site year and geolocation (if MAP data unavailable endemicity as stated in study). ICCs and 95% reference ranges were estimated for country-, study-, and slope-specific heterogeneity (where appropriate) using estimated model variance components.

## Statistical notation for the generalised linear multilevel model (mixed effects, logistic) used to estimate the association between *An. gambiae* gSG6 IgG seropositivity and HBR

The model can be formally written as

$$\text{logit}\left\{\Pr(y_{ij}=1) \mid x_{ij}, \ \zeta_{1j}, \zeta_{2i}, \ \zeta_{3j}\log(HBR)_{ij}\right\} = \beta_1 + \beta_2\log(HBR)_{ij} + \zeta_{1j} + \zeta_{2i} + \zeta_{3i}\log(HBR)_{ij} \quad (1)$$

where

$\zeta_{1j} \sim N(0, \psi_1)$, $\zeta_{2i} \sim N(0, \psi_2)$ and $\zeta_{3j}\log(HBR)_{ij} \sim N(0, \psi_3)$, (2)

where $x_{ij}$ is the vector of model covariates, $\beta_1$ is the model constant and represents the log odds (probability) of gSG6 IgG seropositivity for a log HBR of zero, $\beta_2$ is the fixed effect for log HBR for country $j$ and study $i$, $\zeta_{1j}$ is the random effect (i.e. intercept) for between-country heterogeneity in

probability of gSG6 IgG seropositivity, $\zeta_{2i}$ is the random effect (i.e. intercept) for between-study heterogeneity in probability of gSG6 IgG seropositivity, and $\zeta_{3i}$ is the random effect (i.e. coefficient) for between-study heterogeneity in the effect of log HBR.

## Risk of bias in individual studies

For cross-sectional, cohort or intervention studies, selection bias was assessed by reviewing the studies' inclusion and exclusion criteria. Any case–control studies or studies that presented salivary antibody data stratified by malaria infection status were included in narrative terms only. Risk of bias was assessed by one reviewer using the Risk of Bias in Prevalence Studies tool (*Hoy et al., 2012*). The risk of bias pertains to the reported observations of anti-*Anopheles* salivary antibody seroprevalence included in the multilevel modelling.

## Appendix 2

### Risk of bias assessment

Risk of bias was assessed for each study by one independent reviewer using the Risk of Bias in Prevalence Studies tool (*Hoy et al., 2012*). This tool comprises 10 items and a summary assessment to assess the external validity (selection and non-response bias) and internal validity (measurement bias) of the study's seroprevalence observations. The risk of bias pertains to the reported observations of anti-*Anopheles* salivary antibody seroprevalence included in the multilevel modelling.

With regard to external validity, seven of the studies included in the review were performed in specific populations (i.e. children only) that were not representative of the national population and were deemed to be at high risk of selection bias. Only seven studies included some form of random sampling, and frequently insufficient detail was provided on the sampling frame; as such most studies were included as high risk of selection bias. Furthermore, no studies reported participant response rate, and as such were indicated as high risk of non-response bias.

In terms of internal validity, all studies had an acceptable case definition, with the same mode of data collection, a valid instrument, and an acceptable prevalence period, so were all deemed to be of low risk. However, 12 studies did not include a denominator, instead only reporting the study sample size and prevalence estimate, and were included as high risk.

Overall, due to the specific nature of some of the sample populations for which these prevalence observations are given (i.e. children only) and as participant non-response rate is not given, we conclude that there is a moderate risk of study bias. According to the Risk of Bias in Prevalence Studies tool (*Hoy et al., 2012*), this implies that future research is likely to have an impact on our confidence in the prevalence observations.

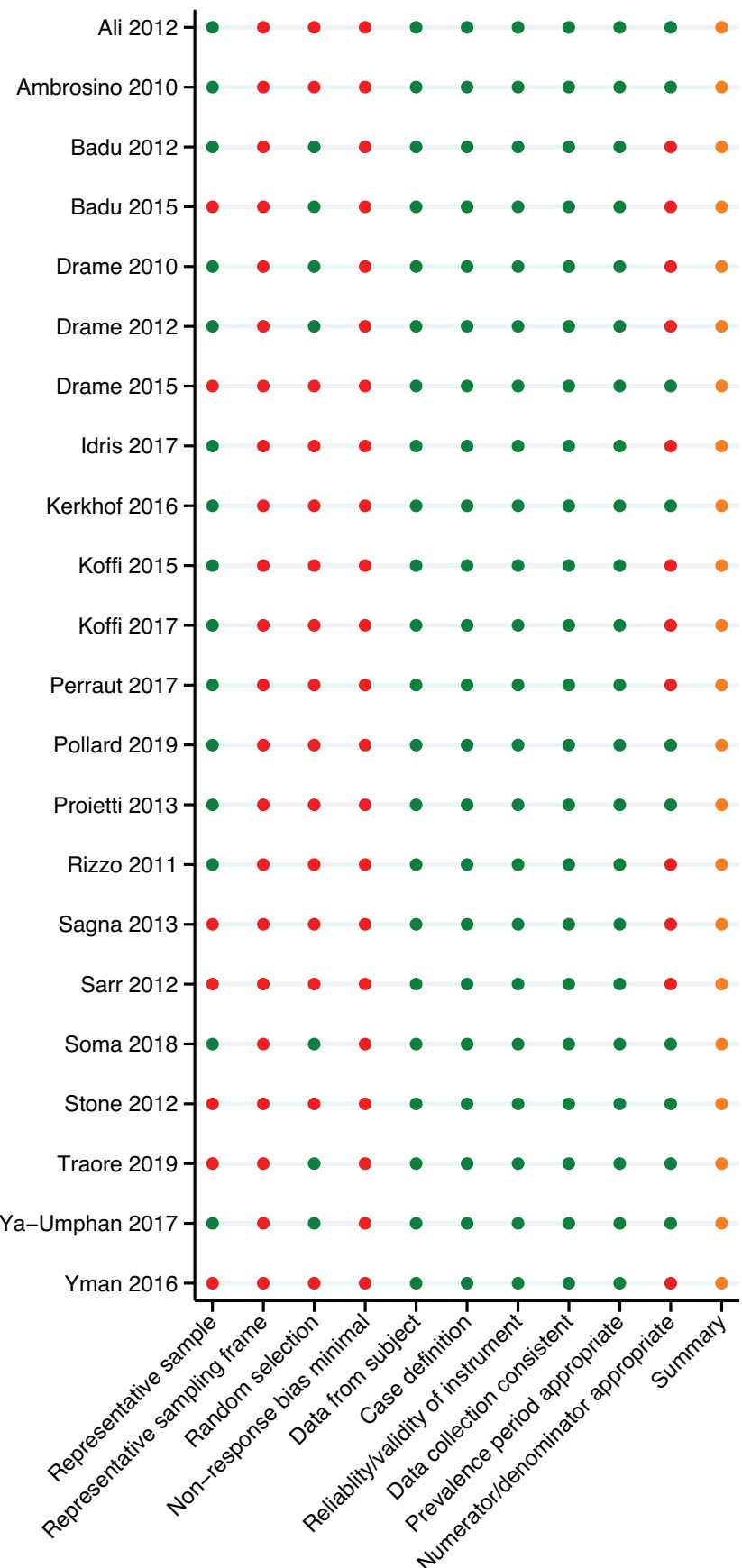

**Appendix 2—figure 1.** Risk of bias assessment. Red, high risk; orange, moderate risk; green, low risk.

# Appendix 3

## Reasons for study exclusion

**Appendix 3—table 1.** Reasons for study exclusion.

| Studies | Reason | References |
|---|---|---|
| 30 | Does not measure anti-salivary antibody responses in individuals/populations | *Arcà et al., 2017*; *Alvarenga et al., 2010*; *Arcà et al., 2005*; *Calvo et al., 2007*; *Calvo et al., 2006*; *Choumet et al., 2007*; *Das et al., 2010*; *Di Gaetano et al., 2018*; *Dixit et al., 2009*; *Francischetti et al., 2014*; *Francischetti et al., 2002*; *Ghosh et al., 2009*; *Isaacs et al., 2018*; *Jariyapan et al., 2010*; *Jariyapan et al., 2006*; *Jariyapan et al., 2012*; *Kamiya et al., 2017*; *Khaireh et al., 2012*; *Korochkina et al., 2006*; *Lombardo et al., 2009*; *Pandey et al., 2018*; *Pedro and Sallum, 2009*; *Phattanawiboon et al., 2016*; *Pirone et al., 2017*; *Rawal et al., 2016*; *Ronca et al., 2012*; *Sarr et al., 2007*; *Scarpassa et al., 2019*; *Wells and Andrew, 2015*; *Zocevic et al., 2013* |
| 28 | Review article | *malERA Refresh Consultative Panel on Tools for Malaria Elimination, 2017*; *malERA Refresh Consultative Panel on Basic Science and Enabling Technologies, 2017*; *malERA Refresh Consultative Panel on Characterising the Reservoir and Measuring Transmission, 2017*; *Andrade and Barral-Netto, 2011*; *Andrade et al., 2005*; *Billingsley et al., 2006*; *Cantillo et al., 2014*; *Coutinho-Abreu et al., 2015*; *Domingos et al., 2017*; *Doucoure et al., 2015*; *Doucoure and Drame, 2015*; *Drame et al., 2013b*; *Fontaine et al., 2011a*; *Foy et al., 2002*; *Gillespie et al., 2000*; *Hopp and Sinnis, 2015*; *Hugo and Birrell, 2018*; *Leitner et al., 2011*; *Lombardo et al., 2006*; *Mathema and Na-Bangchang, 2015*; *Peng et al., 2007*; *Ribeiro and Francischetti, 2003*; *Sagna et al., 2017*; *Sá-Nunes and De Oliveira, 2011*; *Miot and Lima, 2014*; *Peng and Simons, 2004*; *Pingen et al., 2017*; *Sinden et al., 2012* |

*Appendix 3—table 1 Continued on next page*

*Appendix 3—table 1 Continued*

| Studies | Reason | References |
|---|---|---|
| 20 | *Anopheles* salivary antigens not assessed | *Abonuusum et al., 2011*; *Badu et al., 2012a*; *Chaccour et al., 2013*; *Coulibaly et al., 2017*; *Dhawan et al., 2017*; *Fontaine et al., 2011c*; *Fontaine et al., 2011b*; *Jeon et al., 2001*; *Kelly-Hope and McKenzie, 2009*; *Kusi et al., 2014*; *Li et al., 2005*; *Londono-Renteria et al., 2015b*; *Mwanziva et al., 2011*; *Sarr et al., 2011*; *Satoguina et al., 2009*; *Smithuis et al., 2013*; *Ubillos et al., 2018*; *van den Hoogen et al., 2020*; *Varela et al., 2020*; *Wanjala and Kweka, 2016* |
| 10 | Wrong antibody detection methodologies | *Armiyanti et al., 2016*; *Brummer-Korvenkontio et al., 1997*; *Cornelie et al., 2007*; *Fontaine et al., 2012*; *Marie et al., 2014*; *Owhashi et al., 2008*; *Penneys et al., 1989*; *Sor-suwan et al., 2014*; *Sor-Suwan et al., 2013*; *Peng et al., 1998* |
| 7 | Grey literature | *Cornelie et al., 2008*; *Drame et al., 2008*; *Drame et al., 2010c*; *Poinsignon et al., 2008b*; *Poinsignon et al., 2013*; *Poinsignon et al., 2010a*; *Sagna et al., 2018* |
| 6 | Not performed in humans | *Dragovic et al., 2018*; *King et al., 2011*; *Vogt et al., 2018*; *Wang et al., 2013*; *Almeida and Billingsley, 1999*; *Boulanger et al., 2011* |
| 4 | Data already captured by our review from another publication | *Kerkhof et al., 2015*; *Sagna et al., 2013a*; *Ya-Umphan et al., 2018*; *Aka et al., 2020* |
| 3 | Unable to determine appropriate exposure estimate | *Drame et al., 2013a*; *Noukpo et al., 2016*; *Londono-Renteria et al., 2010* |
| 3 | Not in population with natural exposure | *Manning et al., 2020*; *Mendes-Sousa et al., 2018*; *Peng et al., 2004* |
| 1 | Hypothesis study | *Londono-Renteria et al., 2016* |
| 1 | Pooled sera | *Owhashi et al., 2001* |
| 1 | Does not provide estimate of seroprevalence/total levels of antibodies against salivary proteins | *Armiyanti et al., 2015* |
| 1 | Study population not representative: mosquito-allergic patients | *Opasawatchai et al., 2020* |
| 1 | Study population not representative: soldiers with transient exposure | *Orlandi-Pradines et al., 2007* |

# Appendix 4

## Association between gSG6 IgG seropositivity and HBR

**Appendix 4—table 1.** Unadjusted association between gSG6 IgG seropositivity and log human biting rate (HBR).

| Variable | log odds ratio (SE) | 95% CI | p-Value | RE |
|---|---|---|---|---|
| *Fixed part* | | | | |
| log HBR*,† | 0.29 (0.08) | 0.14–0.45 | <0.001 | |
| *Random part* | | | | |
| $\psi_1$‡ | | | | 1.29 |
| $\psi_2$ | | | | 1.55 |
| $\psi_3$ | | | | 0.06 |
| $\rho_1$§ | | | | 0.21 |
| $\rho_2$¶ | | | | 0.47 |
| $\ell$ | | | | −1492.8 |
| *Model fit indices* | | | | |
| Akaike's information criterion | | | | 2995.7 |
| Bayesian information criterion | | | | 3010.1 |

HBR association: log odds ratio and standard error (SE), 95% confidence interval (95% CI), p-value, random-effect components (RE): variances ($\phi$), conditional intraclass correlation coefficient ICC ($\rho$),* and model log likelihood ($\ell$) from generalised linear multilevel modelling (mixed effects, logistic).† This analysis is based upon n = 132 study-specific observations from 12 studies. Of note, five studies that measured HBR and IgG antibodies to gSG6 were excluded from this analysis as they only reported gSG6 IgG levels.

*$\rho = \frac{\psi_k + ... + \psi_{nk}}{\psi_k + ... + \psi_{nk} + \pi^2/3}$ , where $\psi_k$ through $\psi_{nk}$ are random-effect variance estimates pertaining to each of the respective variance components (see table notes ‡–¶) from the generalised linear multilevel modelling (mixed effects, logistic) for a specific ICC estimate.

†Generalised linear multilevel modelling (mixed effects, logistic) estimating the association between log transformed HBR and anti-gSG6 IgG seropositivity with random effects for country-specific and study-specific heterogeneity in gSG6 IgG seroprevalence and study-specific heterogeneity in effect of HBR.

‡$\psi_1$, $\psi_2$, and $\psi_3$ represent variances of the random effects for country, study, and effect of HBR, respectively.

§$\rho_1$ represents conditional ICC for salivary antibody observations from the same country but different study.

¶$\rho_2$ represents conditional ICC for salivary antibody observations from the same country and study with the median HBR.

# Appendix 5

## Association between gSG6 IgG seropositivity and HBR, moderated by dominant vector species

**Appendix 5—table 1.** Association between gSG6 IgG seropositivity and log human biting rate (HBR), moderated by dominant vector species.

| Variable | log odds ratio (SE) | 95% CI | p-Value | RE |
|---|---|---|---|---|
| *Fixed part* | | | | |
| log HBR | 0.46 (0.11) | 0.25–0.66 | <0.001 | |
| DVS | | | <0.001** | |
| *An. gambiae* s.l. only | Ref. | | | |
| *An. gambiae* s.l. and other DVS | 1.00 (0.18) | 0.65–1.25 | <0.001 | |
| Non-*An. gambiae* s.l. | 1.09 (0.68) | –0.24 to 2.42 | 0.109 | |
| log HBR by DVS | | | <0.001** | |
| *An. gambiae* s.l. only | Ref. | | | |
| *An. gambiae* s.l. and other DVS | –0.26 (0.08) | –0.41 to –0.11 | 0.001 | |
| Non-*An. gambiae* s.l. | –0.38 (0.11) | –0.59 to –0.17 | <0.001 | |
| *Random part* | | | | |
| $\psi_1^{\ddagger}$ | | | | 0.96 |
| $\psi_2$ | | | | 2.32 |
| $\psi_3$ | | | | 0.08 |
| $\rho_1^{\S}$ | | | | 0.14 |
| $\rho_2^{\P}$ | | | | 0.51 |
| $\ell$ | | | | –1488.8 |
| *Model fit indices* | | | | |
| Akaike's information criterion | | | | 2995.5 |
| Bayesian information criterion | | | | 3021.5 |

HBR × dominant vector species (DVS) association: log odds ratio and standard error (SE), 95% confidence interval (95% CI), p-value, random-effect components (RE): variances ($\phi$), conditional intraclass correlation coefficient (ICC) ($\rho$),* and model log likelihood ($\ell$) from generalised linear multilevel modelling (mixed effects, logistic).[†] This analysis is based upon n = 132 study-specific observations from 12 studies. Of note, five studies that measured HBR and IgG antibodies to gSG6 were excluded from this analysis as they only reported gSG6 IgG levels.

* $\rho = \frac{\psi_k + ... + \psi_{nk}}{\psi_k + ... + \psi_{nk} + \pi^2/3}$ , where $\psi_k$ through $\psi_{nk}$ are random-effect variance estimates pertaining to each of the respective variance components (see table notes [‡–¶]) from the generalised linear multilevel modelling (mixed effects, logistic) for a specific ICC estimate.

[†] Generalised linear multilevel modelling (mixed effects, logistic) estimating the association between log transformed HBR and anti-gSG6 IgG seropositivity including an interaction term between DVS and log HBR with random effects for country-specific and study-specific heterogeneity in gSG6 IgG seroprevalence and study-specific heterogeneity in effect of HBR.

[‡] $\psi_1$ , $\psi_2$, and $\psi_3$ represent variances of the random effects for country, study, and effect of HBR, respectively.

[§] $\rho_1$ represents the conditional ICC for salivary antibody observations from the same country but different study.

[¶] $\rho_2$ represents the conditional ICC for salivary antibody observations from the same country and study with the median HBR.

**indicates p-value from joint Wald test for polytomous variables.

# Appendix 6

## Association between gSG6 IgG seropositivity and EIR

**Appendix 6—table 1.** Unadjusted association between gSG6 IgG seropositivity log entomological inoculation rate (EIR).

| Variable | log odds ratio (SE) | 95% CI | p-Value | RE |
|---|---|---|---|---|
| *Fixed part* | | | | |
| log EIR | 0.15 (0.04) | 0.07–0.23 | <0.001 | |
| *Random part* | | | | |
| $\psi_1$[‡] | | | | 1.02 |
| $\psi_2$ | | | | 2.15 |
| $\psi_3$ | | | | 0.01 |
| $\rho_1$[§] | | | | 0.16 |
| $\rho_2$[¶] | | | | 0.49 |
| $\ell$ | | | | −530.7 |
| *Model fit indices* | | | | |
| Akaike's information criterion | | | | 1071.3 |
| Bayesian information criterion | | | | 1079.5 |

Entomological inoculation rate (EIR) association: log odds ratio and standard error (SE), 95% confidence interval (95% CI), p-value, random-effect components (RE): variances ($\phi$), conditional intraclass correlation coefficient (ICC) ($\rho$)* and model log likelihood ($\ell$) from generalised linear multilevel modelling (mixed effects, logistic).[†] This analysis is based upon n = 38 study-specific observations from eight studies.

* $\rho = \frac{\psi_k + ... + \psi_{nk}}{\psi_k + ... + \psi_{nk} + \pi^2/3}$, where $\psi_k$ through $\psi_{nk}$ are random-effect variance estimates pertaining to each of the respective variance components (see table notes [‡–¶]) from the generalised linear multilevel (mixed effects, logistic) modelling for a specific ICC estimate.

[†] Generalised linear multilevel modelling (mixed effects, logistic) estimating the association between log transformed EIR and anti-gSG6 IgG seropositivity with random effects for country-specific and study-specific heterogeneity in gSG6 IgG seroprevalence and study-specific heterogeneity in effect of EIR.

[‡] $\psi_1$, $\psi_2$, and $\psi_3$ represent variances of the random effects for country, study, and effect of EIR, respectively.

[§] $\rho_1$ represents the conditional ICC for salivary antibody observations from the same country but different study.

[¶] $\rho_2$ represents the conditional ICC for salivary antibody observations from the same country and study with the median EIR

# Appendix 7

## Association between gSG6 IgG seropositivity and malaria prevalence

**Appendix 7—table 1.** Unadjusted association between gSG6 IgG seropositivity and log *Plasmodium* spp prevalence.

| Variable | log odds ratio (SE) | 95% CI | p-Value | RE |
|---|---|---|---|---|
| *Fixed part* | | | | |
| log *Plasmodium* spp. prevalence | 0.46 (0.32) | –0.16–1.08 | 0.148 | |
| *Random part* | | | | |
| $\psi_1$[‡] | | | | 17.21 |
| $\psi_2$ | | | | 1.25 |
| $\rho_1$[§] | | | | 0.85 |
| $\ell$ | | | | –2597.2 |
| *Model fit indices* | | | | |
| Akaike's information criterion | | | | 5202.5 |
| Bayesian information criterion | | | | 5215.9 |

Any *Plasmodium* species infections (including prevalence estimates of *P. falciparum* only, *P. vivax* only, both *P. falciparum* and *P. vivax* and untyped infections): log odds ratio and standard error (SE), 95% confidence interval (95% CI), p-value, random-effect components (RE): variances ($\phi$), conditional intraclass correlation coefficient (ICC) ($\rho$),* and model log likelihood ($\ell$) from generalised linear multilevel modelling (mixed effects, logistic).[†] This analysis is based upon n = 212 study-specific observations from 14 studies. Of note, six studies that measured *Plasmodium* spp. prevalence and IgG antibodies to gSG6 were excluded from this analysis as five only reported gSG6 IgG levels and one was a case–control study.

* $\rho = \frac{\psi_k + ... + \psi_{nk}}{\psi_k + ... + \psi_{nk} + \pi^2/3}$, where $\psi_k$ through $\psi_{nk}$ are random-effect variance estimates pertaining to each of the respective variance components (see table notes [‡ and §]) from the generalised linear multilevel modelling (mixed effects, logistic) for a specific ICC estimate.

[†] Generalised linear multilevel modelling (mixed effects, logistic) estimating the association between the log prevalence of any *Plasmodium* spp. infection and anti-gSG6 IgG seropositivity with random effects for study-specific heterogeneity in gSG6 IgG seroprevalence and study-specific heterogeneity in effect of *Plasmodium* spp. prevalence.

[‡] $\psi_1$ and $\psi_2$ represent variances of the random effects for study and effect of *Plasmodium* spp. prevalence, respectively.

[§] $\rho_1$ represents the conditional ICC for salivary antibody observations from the same study and with the median *Plasmodium* spp. prevalence.

## Appendix 8

### Association between gSG6 IgG seropositivity and antimalarial antibody seroprevalence

#### Antibodies against *P. falciparum* pre-erythrocytic stage antigens

The pooled analysis of 159 study-specific observations from eight studies showed that a twofold increase in PfCSP IgG seropositivity was associated with a 2.19-fold (OR: 2.19; 95% CI: 1.18–4.04; p=0.013) increase in odds of anti-gSG6 IgG seropositivity (*Stone et al., 2012*; *Ya-Umphan et al., 2017*; *Koffi et al., 2015*; *Koffi et al., 2017*; *Ambrosino et al., 2010*; *Perraut et al., 2017*; *Proietti et al., 2013*; *Kerkhof et al., 2016*). Furthermore, we observed that gSG6 IgG levels increased with increasing PfCSP IgG seroprevalence in four studies (*Ya-Umphan et al., 2017*; *Koffi et al., 2015*; *Koffi et al., 2017*; *Kerkhof et al., 2016*), with another study contributing only a single estimate (*Stone et al., 2012*).

#### Antibodies against *P. falciparum* blood stage antigens

Furthermore, we observed a twofold increase PfAMA1 IgG seroprevalence was associated with a 2.47-fold (OR: 2.47; 95% CI: 2.25–2.71; p<0.001) increase in odds of gSG6 IgG seropositivity based upon 62 study-specific observations from eight studies (*Stone et al., 2012*; *Ya-Umphan et al., 2017*; *Koffi et al., 2015*; *Koffi et al., 2017*; *Perraut et al., 2017*; *Yman et al., 2016*; *Proietti et al., 2013*; *Idris et al., 2017*). A similar association was observed for $PfMSP1_{19}$ IgG, with twofold increase in seroprevalence associated with 2.49-fold (OR: 2.49; 95% CI: 1.21–5.12; p=0.014) increase in odds of gSG6 IgG seropositivity. This association was derived from 163 study-specific observations from 10 studies (*Stone et al., 2012*; *Ya-Umphan et al., 2017*; *Koffi et al., 2015*; *Koffi et al., 2017*; *Perraut et al., 2017*; *Proietti et al., 2013*; *Yman et al., 2016*; *Badu et al., 2015*; *Kerkhof et al., 2016*; *Idris et al., 2017*). Analysis of 47 study-specific observations from three studies indicated that a twofold increase in PfMSP2 IgG seroprevalence was associated with a 41% (OR: 1.41; 95% CI: 1.21–1.65; p<0.001) increase in odds of gSG6 IgG seropositivity (*Ya-Umphan et al., 2017*; *Perraut et al., 2017*; *Yman et al., 2016*). While 17 study-specific observations from two studies showed a twofold increase in PfMSP3 IgG seroprevalence was associated with a 2.66-fold (OR: 2.66; 95% CI: 2.36–3.00; p<0.001) increase in odds of gSG6 IgG seropositivity (*Stone et al., 2012*; *Yman et al., 2016*).

The pooled analysis of 128 study-specific observations from five studies showed that a twofold increase in PfGLURP IgG seroprevalence was associated with a 3.05-fold (OR: 3.05; 95% CI: 2.58–3.61; p<0.001) increase in odds of gSG6 IgG seropositivity (*Koffi et al., 2015*; *Koffi et al., 2017*; *Ambrosino et al., 2010*; *Perraut et al., 2017*; *Kerkhof et al., 2016*). And 18 study-specific observations from five studies indicated that a twofold increase in *P. falciparum* schizont extract IgG seropositivity was associated with a 5.69-fold (OR: 5.69; 95% CI: 0.03–1188.69; p=0.523) increase in odds of gSG6 IgG seropositivity (*Idris et al., 2017*; *Koffi et al., 2015*; *Sarr et al., 2012*; *Perraut et al., 2017*; *Koffi et al., 2017*).

We observed that increasing seroprevalence of IgG antibodies against PfAMA1 saw increased levels of anti-gSG6 IgG in three studies (*Idris et al., 2017*; *Koffi et al., 2015*; *Koffi et al., 2017*), but no association in another (*Ya-Umphan et al., 2017*). The levels of gSG6 IgG increased with increasing $PfMSP1_{19}$ IgG seroprevalence in three studies (*Idris et al., 2017*; *Koffi et al., 2015*; *Badu et al., 2015*), but showed no association in three other studies (*Ya-Umphan et al., 2017*; *Koffi et al., 2017*; *Kerkhof et al., 2016*). No association between gSG6 IgG levels and MSP2 IgG seroprevalence was observed in one study (*Ya-Umphan et al., 2017*). PfGLURP IgG seroprevalence and gSG6 IgG antibody levels were reported in three studies, with one study reporting increased levels (*Koffi et al., 2015*), one study reporting no association (*Kerkhof et al., 2016*), and one study reporting decreased levels of anti-gSG6 IgG with increasing anti-PfGLURP seroprevalence (*Koffi et al., 2017*). One study showed increasing gSG6 IgG levels with increasing *P. falciparum* schizont extract IgG, while three other studies showed no association (*Koffi et al., 2015*; *Koffi et al., 2017*; *Sarr et al., 2012*). Of note, one study provided a single seroprevalence estimate of antibodies against PfAMA1, $PfMSP1_{19}$, and PfMSP3 so no relationships can be drawn (*Stone et al., 2012*).

#### Antibodies against *P. vivax* antigens

In pooled analyses of 115 study-specific observations from two studies (*Idris et al., 2017*; *Kerkhof et al., 2016*), we observed that a twofold increase in the seroprevalence of PvAMA1 was associated with a 3.87-fold (OR: 3.87; 95% CI: 3.46–4.32; p<0.001) increase in the odds of anti-gSG6 IgG

seropositivity. Furthermore, in 103 study-specific observations from two studies (*Idris et al., 2017*; *Kerkhof et al., 2016*), a twofold increase in PvMSP1$_{19}$ IgG seroprevalence was associated with a 2.37-fold (OR: 2.37; 95% CI: 2.26–2.50; p<0.001) increase in the odds of anti-gSG6 IgG seropositivity. However, neither study showed an association between the levels of gSG6 IgG and the seroprevalence of PvAMA1 and PvMSP1$_{19}$ IgG (*Idris et al., 2017*; *Kerkhof et al., 2016*).

**Appendix 8—table 1.** Associations between anti-gSG6 IgG seropositivity and log of antimalarial antibody seroprevalence.

| Exposure | log odds ratio (SE) | 95% CI | p-Value | Study-specific n | Studies | References |
|---|---|---|---|---|---|---|
| *Pre-erythrocytic antigens* | | | | | | |
| log PfCSP IgG seroprevalence (%) | 1.13 (0.45) | 0.24–2.01 | 0.013 | 159 | 8 | *Stone et al., 2012*; *Ya-Umphan et al., 2017*; *Koffi et al., 2015*; *Koffi et al., 2017*; *Ambrosino et al., 2010*; *Perraut et al., 2017*; *Proietti et al., 2013*; *Kerkhof et al., 2016* |
| *Blood stage antigens* | | | | | | |
| log PfAMA1 IgG seroprevalence (%)* | 1.30 (0.07) | 1.17–1.44 | <0.001 | 62 | 8 | *Stone et al., 2012*; *Ya-Umphan et al., 2017*; *Idris et al., 2017*; *Koffi et al., 2015*; *Koffi et al., 2017*; *Perraut et al., 2017*; *Yman et al., 2016*; *Proietti et al., 2013* |
| log PfMSP1$_{19}$ IgG seroprevalence (%) | 1.31 (0.53) | 0.27–2.36 | 0.014 | 163 | 10 | *Stone et al., 2012*; *Ya-Umphan et al., 2017*; *Idris et al., 2017*; *Koffi et al., 2015*; *Koffi et al., 2017*; *Badu et al., 2015*; *Perraut et al., 2017*; *Proietti et al., 2013*; *Yman et al., 2016*; *Kerkhof et al., 2016* |
| log PfMSP2 IgG seroprevalence (%) | 0.50 (0.11) | 0.27–0.72 | <0.001 | 47 | 3 | *Ya-Umphan et al., 2017*; *Perraut et al., 2017*; *Yman et al., 2016* |
| log PfMSP3 IgG seroprevalence (%)* | 1.41 (0.09) | 1.24–1.58 | <0.001 | 17 | 2 | *Stone et al., 2012*; *Yman et al., 2016* |
| log PfGLURP IgG seroprevalence (%) | 1.61 (0.12) | 1.37–1.85 | <0.001 | 128 | 5 | *Koffi et al., 2015*; *Koffi et al., 2017*; *Ambrosino et al., 2010*; *Perraut et al., 2017*; *Kerkhof et al., 2016* |
| log PfSchizont extract IgG seroprevalence (%) | 2.51 (3.93) | −5.20 to 10.22 | 0.523 | 18 | 5 | *Idris et al., 2017*; *Koffi et al., 2015*; *Koffi et al., 2017*; *Sarr et al., 2012*; *Perraut et al., 2017* |
| log PvAMA1 IgG seroprevalence (%) | 1.95 (0.08) | 1.79–2.11 | <0.001 | 115 | 2 | *Idris et al., 2017*; *Kerkhof et al., 2016* |
| log PvMSP1$_{19}$ IgG seroprevalence (%) | 1.25 (0.04) | 1.17–1.32 | <0.001 | 103 | 2 | *Idris et al., 2017*; *Kerkhof et al., 2016* |

Effects for each exposure represent separate generalised linear multilevel modelling (mixed effects, logistic) analyses estimating the association between the log of the seroprevalence of antimalarial antibodies and the seroprevalence of anti-gSG6 IgG, with the inclusion of a random intercept for study-specific heterogeneity and a random coefficient to allow the effect of the antimalarial antigen to vary across studies. Table shows log odds ratio and standard error (SE), 95% confidence interval (95% CI), and p-value, number of study-specific salivary antibody observations (Study-specific n) and studies, with associated references. Random effects not shown. Of note, one study that measured antimalarial antibody seroprevalence and IgG antibodies to gSG6 could not be included in analyses as it only reported gSG6 IgG levels.

* Studies did not include a random coefficient (i.e. slope); as empirical support was not shown.

# Appendix 9

## Country and study-specific predicted probability of gSG6 IgG seropositivity

In order to obtain estimates of gSG6 IgG seroprevalence for each country and study, an intercept-only three-level random-effects logistic regression was fitted to 301 study-specific observations from 22 studies. The predicted probability of gSG6 IgG seropositivity was calculated at the country level (*Appendix 9—figure 1*), indicating that the seroprevalence was lowest in the Pacific region (Vanuatu [31%] and Solomon Islands [32%]) and highest in Benin (72%) and Burkina Faso (65%). Furthermore, the predicted probability of gSG6 IgG seropositivity was calculated at the study level (*Appendix 9—figure 2*), indicating that the seroprevalence was lowest in *Ambrosino et al., 2010* (13%) and highest in *Drame et al., 2015* (91%).

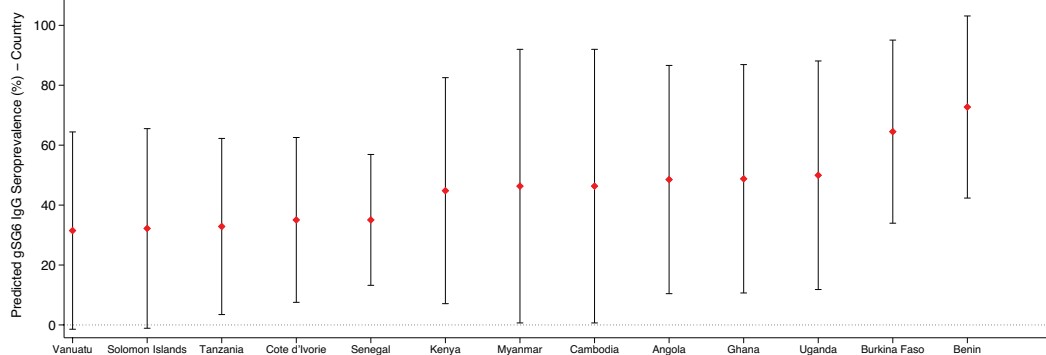

**Appendix 9—figure 1.** Predicted gSG6 IgG seroprevalence by country. Predicted probabilities of gSG6 IgG seropositivity including country-specific random effects with 95% confidence intervals. Estimated from intercept-only three-level random-effects logistic regression to account for the hierarchical nature of the data, with study-specific anti-gSG6 IgG observation nested within study nested within country. Based upon n = 301 study-specific observations from 22 studies. Of note, nine studies that measured IgG antibodies to gSG6 were excluded from this analysis as eight only reported gSG6 IgG levels and one was a case–control study.

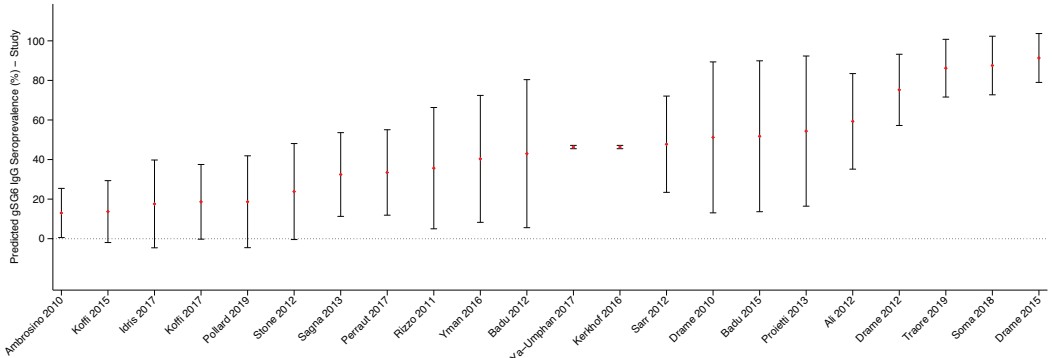

**Appendix 9—figure 2.** Predicted gSG6 IgG seroprevalence by study. Predicted probabilities of gSG6 IgG seropositivity including study-specific random effects with 95% confidence intervals. Estimated from intercept-only three-level random-effects logistic regression to account for the hierarchical nature of the data, with study-specific anti-gSG6 IgG observation nested within study nested within country. Based upon n = 301 study-specific observations from 22 studies. Of note, nine studies that measured IgG antibodies to gSG6 were excluded from this analysis as eight only reported gSG6 IgG levels and one was a case–control study.

# Appendix 10

## Association between alternative salivary biomarkers and exposures of interest

Our systematic review identified a paucity of studies that assessed the relationship between our exposures of interest and most alternate *Anopheles* salivary biomarkers (i.e. non-*An. gambiae* gSG6 IgG), thus preventing the estimation of a pooled association. The exceptions being that we observed that a twofold increase in HBR was associated with a 12% increase (OR: 1.12; 95% CI: 1.02–1.24; p=0.017) in odds of anti-*An. funestus* fSG6 IgG seropositivity (six study-specific observations from two studies; *Rizzo et al., 2011a*; *Ali et al., 2012*; *Appendix 10—table 1*), as well as a 12.97-fold (OR: 12.97; 95% CI: 10.95–15.36; p<0.001) and 4.04-fold (OR: 4.04; 95% CI: 3.60–4.54; p<0.001) increase in odds of anti-gSG6-P2 IgG seropositivity associated with a twofold increase in seroprevalence of PfCSP and PfGLURP IgG, respectively (115 and 116 study-specific observations from two studies, respectively; *Ambrosino et al., 2010*; *Kerkhof et al., 2016*, *Appendix 10—tables 2 and 3*). The associations between exposures of interest and the additional salivary biomarkers are further discussed in narrative terms in below.

**Appendix 10—table 1.** Association between fSG6 IgG seropositivity and human biting rate (HBR).

| Variable | log odds ratio (SE) | 95% CI | p-Value | RE |
|---|---|---|---|---|
| *Fixed part* | | | | |
| log HBR | 0.17 (0.07) | 0.03–0.31 | 0.017 | |
| *Random part* | | | | |
| $\psi_1$‡ | | | | 0.47 |
| $\rho_1$§ | | | | 0.13 |

Association between HBR and fSG6 IgG: log odds ratio and standard error (SE), 95% confidence interval (95% CI), p-value, random-effect components (RE): variances ($\phi$), conditional intraclass correlation coefficient (ICC) ($\phi$)* and model log likelihood ($\ell$) from generalised linear multilevel modelling (mixed effects, logistic).† This analysis is based upon n = 6 study-specific observations.

* $\rho = \frac{\psi_k + ... + \psi_{nk}}{\psi_k + ... + \psi_{nk} + \pi^2/3}$, where $\psi_k$ through $\psi_{nk}$ are random-effect variance estimates pertaining to each of the respective variance components (see table notes ‡ and §) from generalised linear multilevel model (mixed-effects, logistic) for a specific ICC estimate.

† Generalised linear multilevel modelling (mixed effects, logistic) estimating the association between anti-*An. funestus* fSG6 IgG seropositivity and log transformed HBR with random effects for study-specific heterogeneity in fSG6 IgG seropositivity.

‡ $\psi_1$ represents variance of the random effect for study.

§ $\rho_1$ represents conditional ICC for salivary antibody observations from the same study.

### Human biting rate

In addition to the increased odds of *An. funestus* fSG6 seropositivity with increasing HBR, the majority of studies reported a positive association between HBR and the seroprevalence and levels of anti-gSG6-P1 IgM (*Drame et al., 2015*), the levels of gSG6-P2 IgG (*Poinsignon et al., 2008a*), the seroprevalence and levels of anti-cE5 IgG (*Rizzo et al., 2014a*), the levels of anti-fSG6 IgG (*Rizzo et al., 2011a*; *Ali et al., 2012*), the seroprevalence and levels of anti-f5'nuc IgG (*Ali et al., 2012*), and the median levels of anti-*An. gambiae* SGE IgG and IgG4 (*Drame et al., 2010b*; *Lawaly et al., 2012*; *Remoue et al., 2006*). One study reported similar median levels of anti-gSG6 IgG1 across populations and time points, whilst reporting that anti-gSG6 IgG4 titre increased with increasing HBR in one of the populations, but not in the other (*Rizzo et al., 2014b*). Similarly, there was no consistent association between HBR and the levels of anti-cE5 IgG (*Marie et al., 2015*), levels of anti-*An. gambiae* SGE IgE (*Lawaly et al., 2012*) and the seroprevalence and levels of anti-g5'nuc IgG (*Ali et al., 2012*).

### Entomological inoculation rate

*Ali et al., 2012* reported higher seroprevalence and levels anti-fSG6 IgG and anti-f5'nuc IgG with increasing EIR, while anti-g5'nuc IgG seroprevalence and levels were not associated with EIR. An additional study reported gSG6-P2 IgG seroprevalence estimates of 0% for three sites, irrespective of EIR (*Ambrosino et al., 2010*).

## Malaria prevalence

Two studies showed that increased *Plasmodium* spp. prevalence was associated with higher median levels of anti-*An. gambiae* SGE IgG (*Drame et al., 2010b*; *Brosseau et al., 2012*), while another study showed different anti-*An. gambiae* SGE IgG levels for very similar prevalence of malaria and slightly lower levels of anti-*An. gambiae* SGE IgE and IgG4 for the time point with greater malaria prevalence (*Lawaly et al., 2012*). *Kerkhof et al., 2016* showed increasing levels of anti-gSG6-P2 IgG for higher prevalence of any *Plasmodium* spp. infection, while *Londono-Renteria et al., 2020a* showed lower levels of IgG antibodies against TRANS-P1, TRANS-P2, PEROX-P1, PEROX-P2, and PEROX-P3 in the site with higher PCR-confirmed malaria prevalence. Additionally, several case-controlled studies, and two cross-sectional study, reported median antibody levels stratified by malaria infection status. These studies show higher levels of anti-*An. darlingi* SGE IgG (*Andrade et al., 2009*), anti-*An. gambiae* SGE IgG (*Remoue et al., 2006*), anti-*An. dirus* SGE IgG and IgM (*Waitayakul et al., 2006*), and IgG antibodies against SGEs of two Colombian strains of *An. albimanus* in *Plasmodium* spp.-infected individuals compared to non-infected (*Montiel et al., 2020*). While *Montiel et al., 2020* observed no association between anti-*An. darlingi* SGE IgG levels and infection status.

## Antimalarial antibody seroprevalence

Our multilevel modelling indicated that there were 12.97-fold (OR: 12.97; 95% CI: 10.95–15.36; p<0.001) and 4.04-fold (OR: 4.04; 95% CI: 3.60–4.54; p<0.001) increase in odds of anti-gSG6-P2 IgG seropositivity associated with a twofold increase in the seroprevalence of PfCSP and PfGLURP IgG, respectively (*Ambrosino et al., 2010*; *Kerkhof et al., 2016*; *Appendix 10—table 2* and *Appendix 10—table 3*). However, we observed weak positive associations between the levels of IgG antibodies against gSG6-P2 peptide and the seroprevalence of IgG antibodies against PfMSP1$_{19}$, PfGLURP and PvMSP1$_{19}$, but no association with PfCSP or PvAMA1 (*Kerkhof et al., 2016*).

**Appendix 10—table 2.** Association between gSG6-P2 IgG seropositivity and log PfCSP IgG seroprevalence.

| Variable | log odds ratio (SE) | 95% CI | p-Value | RE |
|---|---|---|---|---|
| *Fixed part* | | | | |
| log PfCSP IgG seroprevalence | 3.70 (0.12) | 3.45–3.94 | <0.001 | |
| *Random part* | | | | |
| $\psi_1$‡ | | | | 25.2 |
| $\rho_1$§ | | | | 0.88 |

Association between log PfCSP seroprevalence and gSG6-P2 IgG: log odds ratio and standard error (SE), 95% confidence interval (95% CI), p-value, random-effect variances ($\phi$), conditional intraclass correlation coefficient (ICC) ($\rho$)* and model log likelihood ($\ell$) from logistic mixed-effects modelling.† This analysis is based upon n = 115 study-specific observations.

* $\rho = \frac{\psi_k + ... + \psi_{nk}}{\psi_k + ... + \psi_{nk} + \pi^2/3}$, where $\psi_k$ through $\psi_{nk}$ are random-effect (RE) variance estimates pertaining to each of the respective variance components (see table notes ‡ and §) from generalised linear multilevel model (mixed effects, logistic) for a specific ICC estimate.

† Generalised linear multilevel modelling (mixed effects, logistic) estimating the association between log PfCSP seroprevalence and anti-gSG6-P2 IgG seropositivity with random effects for study-specific heterogeneity in gSG6-P2 IgG seropositivity.

‡ $\psi_1$ represents variance of the random effect for study.

§ $\rho_1$ represents conditional ICC for salivary antibody observations from the same study.

**Appendix 10—table 3.** Association between gSG6-P2 IgG seropositivity and log PfGLURP IgG seroprevalence.

| Variable | log odds ratio (SE) | 95% CI | p-Value | RE |
|---|---|---|---|---|
| *Fixed part* | | | | |
| log PfGLURP IgG seroprevalence | 2.01 (0.09) | 1.85–2.18 | <0.001 | |
| *Random part* | | | | |
| $\psi_1$‡ | | | | 24.3 |

*Appendix 10—table 3 Continued on next page*

*Appendix 10—table 3 Continued*

| Variable | log odds ratio (SE) | 95% CI | p-Value | RE |
|---|---|---|---|---|
| $\rho_1{}^{\S}$ | | | | 0.88 |

Association between log PfGLURP seroprevalence and gSG6-P2 IgG: log odds ratio and standard error (SE), 95% confidence interval (95% CI), p-value, random-effect variances ($\phi$), conditional intraclass correlation coefficient (ICC) ($\rho$)* and model log likelihood ($\ell$) from logistic mixed-effects modelling.[†] This analysis is based upon n = 116 study-specific observations.

*$\rho$ = , where $\phi_k$ through $\phi_{nk}$ are random-effect (RE) variance estimates pertaining to each of the respective variance components (see table notes [‡ and §]) from generalised linear multilevel model (mixed effects, logistic) for a specific ICC estimate.

[†]Generalised linear multilevel modelling (mixed effects, logistic) estimating the association between log PfGLURP seroprevalence and anti-gSG6-P2 IgG seropositivity with random effects for study-specific heterogeneity in gSG6-P2 IgG seropositivity.

[‡] $\phi_1$ represents variance of the random effect for study.

[§] $\rho_1$ represents conditional ICC for salivary antibody observations from the same study.

