## [Editor Report]

We believe this systematic review on the use of serological data to monitor anopheline mosquito exposure will add to the existing literature and help provide important insight into how these markers may be used to understand malaria transmission.

---

## [Decision Letter]

**Decision letter after peer review:**

Thank you for submitting your article "*Anopheles* salivary antigens as serological biomarkers of vector exposure and malaria transmission: A systematic review with multilevel modelling" for consideration by *eLife*. Your article has been reviewed by 2 peer reviewers, and the evaluation has been overseen by a Reviewing Editor and David Serwadda as the Senior Editor. The following individual involved in review of your submission has agreed to reveal their identity: Philip Bejon (Reviewer #2).

Essential revisions:

1. The paper would be strengthened if it focused on Africa, with sub-group analyses between Africa and Asia, given the differences in vector behavior in both settings.

2. The overall conclusions often use phrasing such as 'strong' evidence/association, however in many instances this level of support is overly optimistic, given the noise in these types of data and issues with a number of the malaria metrics. Further, for some of the statistical analyses require additional justification and more robust model evaluation.

3. There are a number of points raised in both reviewers where the results should be clarified (ex. text and display of Figure 2; relationship between HBR and seroprevalence as a metric; etc.). In addition, the sources of data for each point of evidence – is it MAP estimates? Microscopy? Which type of study design? Should be clarified to help provide the reader with a better understanding of the pieces of information (and level of bias, etc.) are providing which types of evidence.

*Reviewer #1:*

This is an interesting paper highlighting the use of serological data for monitoring anopheline mosquito bites. The concept is good, rationale clear and the statistical modelling appears robust. Some of the stats could be better explained at times and the conclusions should be tapered somewhat as there is lots of "strong" evidence/association statements that seem overly optimistic given the noise in these types of data and the frailties of the malaria metrics. This meta-analyses will help crystalise thought on whether the technique should be further invested in.

(1) The study groups together cross-sectional and cohort data yet this was never really explored. How strong was the association in the cohort data and did it give the same trends? This is important if recommending using this method for surveillance as there is no mention of antibody decay.

(2) Abstract and through paper – not clear what a "meta-observation" is. Please clarify.

(3) Figure 2 could be improved. It plots the x-axis on the normal scale when it was fitted on the log. Statements about how it could be used in elimination settings were given but we cannot see how well the model behaves down this region. Consider putting on the log. Given (2) not clear what the size of the points are.

(4) Model fitting – a ln transformation was used which could do with a justification. It could be fine, but we cannot tell given the plotting outlined above. Given the choice of transformation determines values presented in figure 3 I would like to see either a more rigorous model selection or at least a plot which indicates this is a reasonable choice. This could be done by binning the data and including a repeat of Figure 2 and 5 with these binned data in the appendix.

(5) Table 1 is unwieldly and hard to understand what the breakdown of the different metrics are. This would benefit from either a summary table/figure or a narrative in the results (i.e. what % of studies/datapoints had HBR, EIR etc). What % datapoints were from longitudinal studies?

(6) Why is the figure for the relationship between sero-prevalence and EIR not shown, though model predictions are? Should this not go in the main text.

(7) The malaria data is a hodgepodge of microscopy, PBR and MAP estimated values. This is fine, though clearly some of these are more bias than others. The problem is that it isn't really differentiated in the Results section leaving me unsure of whether it was the data or the MAP predictions that were finding the association. Specifically, did the microscopy or PCR data predict a positive association alone or was it the modelled estimates which generate the positive association. Clearly, if it is the former, the story is stronger but given the narrative in lines 254-258 when 8 studies did not find an association, I am unsure. Could the points be coloured by the source of the prevalence estimate in Figure 5?

(8) The authors are conflating seropositivity for anopheles bites with malaria transmission. There are many reasons why this might not be so, but the most obvious is altitude and use of vector control. Altitude could explain why 20% people have mosquito bites but <1% malaria and given you have geospatial information this could be easily tested (ie. Excluding sites over 2000m). The use of control interventions/treatment is harder to control for and could be discussed more thoroughly in light of comment (1)

(9) Finally, though the analyses is interesting, the frailties of using geospatial data should be discussed given that this is often based on a paucity of data.

*Reviewer #2:*

The problem to be addressed is well formulated, and there are sufficient secondary data available to conduct a useful analysis to address it. The statistical approach is careful and multi-level modelling is an appropriate way of dealing with the linked observations in this dataset.

I thought a stronger case would be made if the paper focused on Africa, as vector behaviour is so very different in Asia, and I would like to see a sub-group analysis of the two. There is good evidence for a non-linear relationship as reported by the authors, and the graph in Figure 2 is persuasive: but I found it difficult to follow the quantification of this in the text (details in private review). It is a matter of preference, but I would relegate more of the material after figure 2 to supplementary figures so as to focus the presentation more on this figure.

The authors are well justified in claiming a relationship between HBR and seroprevalence of mosquito salivary antigen, and this could be useful for vector surveillance. The shape of the line indicates a dynamic range where seroprevalence is a useful metric, and a statement of this range could be added.

In practice, the investment in this form of vector surveillance would need to be balanced against spending in other forms of disease or infection surveillance, and this could be mentioned as an issue in terms of a policy impact.

(1) Abstract: If the association is non-linear is it reasonable to represent it with a % in X leads to % in y? Also multi-level modelling in methods could usefully give an indication of what the levels used were.

(2) Introduction is well formulated and I have no comments

(3) Methods: I agree with the decision to use seropositivity rather than levels, and also to include studies that classify based on seronegative.

(4) Were estimates were taken from MAP or Pangea, at what level was geolocation used? Longitude/latitude or administrative category?

(5) MAP focuses on PfPR, how much data has fed into DVS?

(6) The random intercepts sentence has three levels, but gives same study and same country – please clarify which relates to which level?

(7) I think it would be worth examining Africa and outside-Africa separately, to get a sense of whether the relationships are different. (Or use an interacting term for continent and slope).

(8) The methods section does not help me understand how non-linearity was allowed for. Natural log transformations are mentioned, but if these are applied to both x's and y then we end up linear again. So a combination must have been selected. Was there any testing of whether the non-linear fit was superior to the linear fit?

(9) Results – Figure 1 – I would be interested in knowing how many studies were excluded because seroprevalence was described differently from the parameters set.

(10) Results – again I struggled with the description of non-linearity here. It seems like IgG was kept linear, HBR was log transformed. Therefore shouldn't there be a % increase in the log transformed variable for each unit of absolute increase in the untransformed variable?

(11) Figure 2 – a lot of the "action" is down below 1 HBR. This is expected, and I have no problem with the shape of the fit, but I would be interested in seeing this more clearly in a log-transformed x axis.

(12) Having struggled with the non-linearity up to Figure 3, I couldn't then cope with the increase Figure 3 was communicating. Isn't a percent increase just a linear factor away from an Odds Ratio?

(13) For figures 2 and 5 there is obviously a lot of variation left unexplained judging by the scatter – including some quite large studies. I think this should be commented on in the results and discussion. Did the authors check if the risk of bias tended to be linked with outlying observations?

(14) I felt uncomfortable in combining Asia and Africa.

(15) I thought the discussion was well balanced overall. As above, some discussion on the very wide scatter would be useful, and also the fact that the relationship between seroprevalence and HBR is flat above an HBR of 1.5 limits the usefulness of this method in that dynamic range. The authors could comment on how many malaria endemic regions would be covered in the 0-1.5 range. The authors could also comment on the advantage or complimentarity of malaria antigen serology for surveillance in comparison.

---

## [Author Response]

Essential revisions:1. The paper would be strengthened if it focused on Africa, with sub-group analyses between Africa and Asia, given the differences in vector behavior in both settings.

Our systematic review has demonstrated that gSG6 serology has been used as a marker of *Anopheles* exposure globally and we agree that it is important to take into consideration heterogeneity in vector behaviour. In our manuscript we directly estimate effect modification by dominant vector species (DVS). The three categories of DVS serve as a proxy for region with Africa represented by 2 categories (*An. gambiae s.l.*; *An gambiae s.l.* + others) and other regions represented by the non *An. gambiae* category. We have therefore emphasised this and highlighted the African context throughout the manuscript. We provide further details in response to Reviewer 2’s Comments #7 and #14.

2. The overall conclusions often use phrasing such as 'strong' evidence/association, however in many instances this level of support is overly optimistic, given the noise in these types of data and issues with a number of the malaria metrics. Further, for some of the statistical analyses require additional justification and more robust model evaluation.

We have tempered the language of the conclusions and have provided justification of the model selection process and analyses (see Appendix 1 – Supplementary Methodology).

3. There are a number of points raised in both reviewers where the results should be clarified (ex. text and display of Figure 2; relationship between HBR and seroprevalence as a metric; etc.). In addition, the sources of data for each point of evidence – is it MAP estimates? Microscopy? Which type of study design? Should be clarified to help provide the reader with a better understanding of the pieces of information (and level of bias, etc.) are providing which types of evidence.

We have carefully considered this feedback and have improved the display of Figure 2 and provided a more in-depth narrative description of the study details to assist with interpretation of model parameters. We have also addressed all reviewers’ suggestions and included clarifications on model input where recommended.*Reviewer #1:*

This is an interesting paper highlighting the use of serological data for monitoring anopheline mosquito bites. The concept is good, rationale clear and the statistical modelling appears robust. Some of the stats could be better explained at times and the conclusions should be tapered somewhat as there is lots of "strong" evidence/association statements that seem overly optimistic given the noise in these types of data and the frailties of the malaria metrics. This meta-analyses will help crystalise thought on whether the technique should be further invested in.

We have tempered the conclusions to remove references to “strong” evidence throughout and emphasised where confidence intervals were wide and study-specific heterogeneity was observed. We have also expanded our discussion of the limitations of the malaria metrics (see detailed response to point #9 below). We have already provided a cautious discussion about the limitations regarding the utility of this approach (Discussion, Page 23-25, Line 361-400), highlighting areas for further development and posing several key questions to answer (i.e. appropriateness as a biomarker of non-*An. gambiae*, individual-level exposure and antibody half-life) before it can be used as a tool for population- or individual-level predictions.

(1) The study groups together cross-sectional and cohort data yet this was never really explored. How strong was the association in the cohort data and did it give the same trends? This is important if recommending using this method for surveillance as there is no mention of antibody decay.

We explored the moderating effect of study design (longitudinal cohort versus cross-sectional) on the association between the log HBR and gSG6 IgG seroprevalence and found no significant effect modification by study type (*p*-value for interaction: 0.138).

(Results, Page 14 Line 210-213) “There was no evidence that the association between HBR and gSG6 IgG varied according to vector collection method (human landing catch or other indirect methods**;**
*p*=0.443) or study design (longitudinal cohort or cross-sectional/repeated cross-sectional; *p*=0.138).”

The question of antibody decay is an interesting one, which unfortunately we cannot examine due to the structure of the data. Furthermore, given there are only 3 studies that report gSG6 and HBR longitudinally, we are unlikely to draw any conclusions about antibody decay dynamics beyond which were discussed in the original manuscripts.

This limitation has already been mentioned in the Discussion (Page 25, Line 391-395):

“However, the accuracy of salivary antibodies to measure individual-level exposure to *Anopheles* bites is yet to be validated; literature searches identified no studies investigating this association at the individual-level. Without detailed measurements of individual-level vector exposure, or a detailed knowledge of the half-life of *Anopheles* salivary antibodies post biting event, the true accuracy of salivary antibodies, such as SG6 IgG, to measure individual-level HBR remains unknown.”

(2) Abstract and through paper – Not clear what a "meta-observation" is. Please clarify.

Study reported observations of salivary antibody seroprevalence or levels were extracted at the most granular level for each study (i.e. for each site, time point). Each of these study-specific salivary antibody observations were included as the level-one units in our multilevel model, with study as level-two to account for multiple observations from the same study, and country as level-three to account for multiple studies from the same country.

To clarify, we have renamed meta-observations and Meta-N to be “study-specific observations (n)” in the text and study-specific n in Table 1.

In addition, we have now added to the outcome section of the methods (Page 4-5, Line 95-97):

“Study reported salivary antibody data was extracted at the most granular level (i.e. for each site; time point), with each observation of seroprevalence or levels included as a study-specific salivary antibody observation.”

(3) Figure 2 could be improved. It plots the x-axis on the normal scale when it was fitted on the log. Statements about how it could be used in elimination settings were given but we cannot see how well the model behaves down this region. Consider putting on the log. Given (2) not clear what the size of the points are.

To address both reviewers’ comments on Figure 2, we have modified the presentation and discussion of the results to be for a two-fold increase in HBR associated with an increase in gSG6 IgG seroprevalence. Figure 2 is now shown on the log base 2 scale, but with labels for the equivalent HBR values (e.g. labels 1, 2, 4, 8 for HBR values correspond to 0, 1, 2, 3 on log base 2 scale) to ensure interpretability and clinical relevance. We have also coloured the markers relative to sample size, as outlined in the figure legend.

(Results, Page 14, Line 199-207) “Generalised linear multilevel modelling (mixed-effects, logistic) of n=132 study-specific observations from 12 studies estimated a positive association between *Anopheles* spp.-HBR (log transformed) and seroprevalence of IgG to *An. gambiae* gSG6 salivary antigen [5, 7, 8, 10, 11, 13, 29, 33, 36, 38, 39, 53] (Figure 2 and Appendix 4 – Table 1). […] For example, the magnitude of the association was such that a 2-fold (100% relative) increase in HBR was associated with a 23% increase (OR: 1.23; 95%CI: 1.10-1.37, *p*<0.001) in the odds of anti-gSG6 IgG seropositivity (Figure 2).”

(4) Model fitting – a ln transformation was used which could do with a justification. It could be fine, but we cannot tell given the plotting outlined above. Given the choice of transformation determines values presented in figure 3 I would like to see either a more rigorous model selection or at least a plot which indicates this is a reasonable choice. This could be done by binning the data and including a repeat of Figure 2 and 5 with these binned data in the appendix.

To address suggestions from both reviewers we have added further clarification of the model selection process to the methods and Supplementary Methods to show model selection process.

(Methods, Page 6, Lines 138-143) “For the exposure measures (HBR, EIR, malaria prevalence and antimalarial antibody seroprevalence), the data were log transformed since there were non-linear associations between the exposure measures on the original scale and seroprevalence – supported empirically by superior model fit as indicated by Akaike's information criterion (AIC) and Bayesian information criterion (BIC) fit indices (Appendix 1 – Table 1). To aid interpretation, we present our results as a relative increase in the odds of the gSG6 IgG seropositivity for a 2-fold or in other words a 100% relative increase in the exposures.”

(Appendix 1, Page 43, Lines 1161-1166) “Akaike's information criterion (AIC) and Bayesian information criterion (BIC) fit indices were used to determine the best fitting functional forms for the association between log odds of gSG6 IgG seropositivity and HBR, EIR and *Plasmodium* spp. prevalence – linear, log, quadratic and cubic functions were fitted, with a log transformation exhibiting superior model fit (Appendix 1 – Table 1).”

(5) Table 1 is unwieldly and hard to understand what the breakdown of the different metrics are. This would benefit from either a summary table/figure or a narrative in the results (i.e. what % of studies/datapoints had HBR, EIR etc). What % datapoints were from longitudinal studies?

We have chosen to further discuss these metrics in the first paragraph of the Results section.

(Results, Page 7-8, Lines 156-185) “Literature searches identified 158 potentially relevant studies, of which 42 studies were included in the systematic review (Figure 1) and are described in Table 1. […] The distributions of exposure estimates were: HBR (n=197 from 24 studies, median: 3.0 bites per person per night, IQR: 0.9-12.1; range: 0-121.4), EIR (n=60 from 8 studies, median: 7.3 infectious bites received per person per year, IQR: 0-36.4; range: 0-585.6), and *Plasmodium* spp. Prevalence (n=266 from 22 studies, median: 9.1%; IQR: 4-22%; range: 0-94.6%).”

(6) Why is the figure for the relationship between sero-prevalence and EIR not shown, though model predictions are? Should this not go in the main text.

We have now included a figure showing the relationship between seroprevalence and log_2_ EIR as Figure 4, and discussed this in the main text as a 2-fold increase in EIR.

(Results, Page 17, Lines 246-250) “For **a** 2-fold increase in EIR, the odds of anti-gSG6 IgG seropositivity increased by 11% (OR: 1.11; 95%CI: 1.06-1.17; *p*<0.001), with heterogeneity in the study-specific effects (95% reference range: 1.00-1.24; likelihood ratio χ^2^ (1) = 15.02, *p*<0.001).”

(7) The malaria data is a hodgepodge of microscopy, PBR and MAP estimated values. This is fine, though clearly some of these are more bias than others. The problem is that it isn't really differentiated in the Results section leaving me unsure of whether it was the data or the MAP predictions that were finding the association. Specifically, did the microscopy or PCR data predict a positive association alone or was it the modelled estimates which generate the positive association. Clearly, if it is the former, the story is stronger but given the narrative in lines 254-258 when 8 studies did not find an association, I am unsure. Could the points be coloured by the source of the prevalence estimate in Figure 5?

We have now clarified that the malaria data used in Figure 5 are study measured *Plasmodium* spp. prevalence estimates and not Malaria Atlas Project estimates. In addition, to be consistent, we have amended Figure 5 and the discussion to reflect a 2-fold increase in *Plasmodium* spp. prevalence.

(Results, Page 19, Line 265-271) “Generalised linear multilevel modelling (mixed-effects, logistic) of n=212 from 14 studies that measured *Plasmodium* spp. prevalence contemporaneously in their study [3, 5, 9, 13, 15, 29, 30, 33, 35, 36, 38, 41, 47, 51] showed that for a 2-fold increase in the prevalence of *Plasmodium* spp. infection the odds of gSG6 IgG seropositivity increased by 38%, although the confidence intervals were wide (OR: 1.38; 95%CI: 0.89-2.12; *p*=0.148) and heterogeneity in the study-specific effects was observed (95% reference range: 0.30-6.37; likelihood ratio χ^2^ (1) = 235.5, *p*<0.001) (Figure 5 and Appendix 7).”

(8) The authors are conflating seropositivity for anopheles bites with malaria transmission. There are many reasons why this might not be so, but the most obvious is altitude and use of vector control. Altitude could explain why 20% people have mosquito bites but <1% malaria and given you have geospatial information this could be easily tested (ie. Excluding sites over 2000m). The use of control interventions/treatment is harder to control for and could be discussed more thoroughly in light of comment (1).

In many settings (especially those with low transmission) there would be instances where an individual would receive bites from uninfected mosquitoes, thus why estimates of seropositivity for antibodies against *Anopheles* salivary proteins would likely always be greater than parasite prevalence/endemicity. An analysis restricted due to altitude therefore would not necessarily change this (furthermore, all study sites are below 2000m). Instead, we discuss on Page 24 Lines 369-375 how a salivary antigen approach to measuring malaria transmission is limited in this sense, and would likely need to be combined with another measure (e.g. malaria prevalence, antimalarial antibody seroprevalence) to accurately estimate transmission intensity.

(9) Finally, though the analyses is interesting, the frailties of using geospatial data should be discussed given that this is often based on a paucity of data.We have now included a discussion of the limitations of geospatial data in our discussion.

(Discussion, Page 25-26, Line 415-423) “In light of this, we also chose to provide estimates of association and gSG6 IgG seroprevalence according to a selected range of epidemiologically relevant hypothetical HBR’s (no widely accepted HBR classification exists in the literature) and according to widely accepted, discrete, endemicity classes according to MAP estimates (which permitted inclusion of all studies) to provide epidemiological context. […] Any misclassification events may cause us to underestimate the standard error in the effect of malaria endemicity class and DVS on gSG6 IgG.”

Reviewer #2:(1) Abstract: If the association is non-linear is it reasonable to represent it with a % in X leads to % in y? Also multi-level modelling in methods could usefully give an indication of what the levels used were.

As we have log transformed the exposure variable, using exponentiated coefficients, the results should be interpreted as a relative per cent (i.e. a 100% (or a 2-fold)) increase in the exposure will be associated with a relative per cent increase in the odds of the outcome. However, to avoid confusion we now express it as a 2-fold increase in X leads to a % increase in Y.

We have amended the abstract to indicate model levels.

(Abstract, Page 2, Line 26-29) “Multilevel modelling (to account for multiple study-specific observations (level-one), nested within study (level-two), and study nested within country (level-three)) estimated associations between seroprevalence with Anopheles human biting rate (HBR) and malaria transmission measures.”

(2) Introduction is well formulated and I have no comments,

Noted, with thanks.

(3) Methods: I agree with the decision to use seropositivity rather than levels, and also to include studies that classify based on seronegative.

Noted, with thanks.

(4) Where estimates were taken from MAP or Pangea, at what level was geolocation used? Longitude/latitude or administrative category?

Longitude and Latitude were used. We have amended the Methods to reflect this

(Methods, Page 5, Line 108-112) “Where exposure estimates were not provided, we attempted to source data from other publications by the authors, or using the site geolocation (longitude and latitude) and year to obtain estimates of EIR from the Pangaea dataset [18], *P. falciparum* rates in 2-10 year olds (PfPR2-10) and dominant vector species (DVS) from the Malaria Atlas Project (MAP) [19].”

(5) MAP focuses on PfPR, how much data has fed into DVS?

The DVS occurrence data from the MAP is based upon a collated database of 15,837 vector occurrence data points using species-specific location information from over 4800 sources (collated and reported in Sinka *et al.* [1]). While there are several limitations associated with these maps (including sparse data for some areas, and that at time of use they are 10-years out of date), they are the best currently available indication of the distribution of dominant malaria vectors. We used MAP to inform DVS categories for only 7 (out of 42 studies). To address comments from both reviewers, we have included a brief discussion of the limitations of using MAP data in the discussion.

(Discussion, Page 25-26, Line 415-423) “In light of this, we also chose to provide estimates of association and gSG6 IgG seroprevalence according to a selected range of epidemiologically relevant hypothetical HBR’s (no widely accepted HBR classification exists in the literature) and according to widely accepted, discrete, endemicity classes according to MAP estimates (which permitted inclusion of all studies) to provide epidemiological context. […] Any misclassification events may cause us to underestimate the standard error in the effect of malaria endemicity class and DVS on gSG6 IgG.”

(6) The random intercepts sentence has three levels, but gives same study and same country – please clarify which relates to which level?

Please see (Methods, Page 5-6, Line 122-124) “Random intercepts for study and country were estimated to account for nested dependencies induced from multiple study-specific salivary antibody observations (level-one) from the same study (level-two) and studies from the same country (level-three).

(7) I think it would be worth examining Africa and outside-Africa separately, to get a sense of whether the relationships are different. (Or use an interacting term for continent and slope).

In the manuscript we chose to explore interactions with dominant vector species given the species-specific nature of the antibodies/antigens under investigation. However, region (Africa/non-Africa) is a proxy for dominant vector species – all non-African sites are represented by non-*An. gambiae s.l.*, and all African sites are represented as DVS: *An. gambiae s.l.* only or *An. gambiae s.l.* + others. (Note: while one African study reported *An. funestus* as the only DVS, this study only provided data on antibody levels and was not included in multilevel modelling). We have now emphasised the Africa/non-African context throughout the manuscript with the following notable examples:

(Introduction, Page 3-4, Lines 70-76) “We undertook a systematic review with multilevel modelling, to quantify the association between HBR, EIR, and other markers of malaria transmission, with anti-*Anopheles* salivary antibody responses and to understand how these associations vary according to transmission setting and dominant *Anopheles* vectors which can exhibit different biting behaviours. In particular, we were interested in comparing the African context (where *An. gambiae* and *P. falciparum* predominates) to non-African settings (where *An. gambiae* is absent and where both *P. falciparum* and *P. vivax* are prevalent).”

(Results, Page 7, Lines 159-166) “These studies were performed in 16 countries mostly in hypo or mesoendemic areas of Africa (32 studies), with a minority performed in South America (4 studies), Asia (4 studies), and the Pacific (2 studies). […] *An. gambiae s.l.* was a DVS in all African study sites (n=151 study-specific observations from 23 studies where *An. gambiae s.l.* was the only DVS and n=68 from 16 studies where *An. gambiae s.l.* was present with additional DVS (i.e. *An. funestus*, *An. pharoensis*)), with the exception of one study, which together with the 10 non-African studies contributed n=174 study-specific estimates where *An. gambiae s.l.* was absent.”

(Results, Page 14, Lines 213-226) “Given the global heterogeneity in the distribution of *Anopheles* species, we sought to quantify the extent to which the association between *An. gambiae* gSG6 IgG seropositivity and HBR is moderated by DVS. […] In African studies where *An. gambiae s.l* is the only DVS, predicted seroprevalence of *An. gambiae* gSG6 ranged from 8% (95%CI: 0-22%) to 86% (95%CI: 67-100%) for an HBR of 0.01 to 100 bites per person per night respectively (Figure 3 and Figure 3 – Supplement 1).”

(Results, Page 19, Lines 271-274) “In the association between gSG6 IgG seropositivity and *Plasmodium* spp. infection, there was no evidence for a moderating effect of *Plasmodium* spp. detection method (light microscopy, or PCR, *p*=0.968), or species (African studies with *P. falciparum* versus non-African studies where *P. falciparum* and *P. vivax* are co-prevalent*, p*=0.538).”

(Results, Page 21, 302-303) “Interactions with DVS or region (Africa/non-Africa) could not be explored due to collinearity with malaria endemicity class.”

(8) The methods section does not help me understand how non-linearity was allowed for. Natural log transformations are mentioned, but if these are applied to both x's and y then we end up linear again. So a combination must have been selected. Was there any testing of whether the non-linear fit was superior to the linear fit?

We have updated the text in the methods section to indicate model selection process. Natural log transformations were applied to the exposure data (HBR/EIR/ malaria prevalence and anti-malarial antibody seroprevalence) as the exposure data on the original scale had a non-linear association with the log odds of anti-gSG6 IgG seropositivity. See response to Comment #1 with respect to how exponentiated coefficients are interpreted where effects for a log transformed exposure are estimated in a logit regression model. As per Reviewer 1’s Comment #4, we have also included a more detailed description in Appendix 1 – Supplementary Methodology to demonstrate model selection process.

(Appendix 1, Page 42-43, Lines 1161-1166) “Akaike's information criterion (AIC) and Bayesian information criterion (BIC) fit indices were used to determine the best fitting functional forms for the association between log odds of gSG6 IgG seropositivity and HBR, EIR and *Plasmodium* spp. prevalence – linear, log, quadratic and cubic functions were fitted, with a log transformation exhibiting superior model fit (Appendix 1 – Table 1).”

(9) Results – Figure 1 – I would be interested in knowing how many studies were excluded because seroprevalence was described differently from the parameters set.

The studies that measured antibodies to *Anopheles* salivary proteins, but did not report seroprevalence at all or according to our parameter set, were included in narrative terms as per the study protocol. In total 20 studies were not able to be included in our multilevel modelling analyses: 15 only reported antibody levels, 3 were the only study reporting the seroprevalence of antibodies to that *Anopheles* salivary antigen, and 2 were case-control studies. We have clarified this in the footnotes of the modelling outputs in the various appendices.

(10) Results – again I struggled with the description of non-linearity here. It seems like IgG was kept linear, HBR was log transformed. Therefore shouldn't there be a % increase in the log transformed variable for each unit of absolute increase in the untransformed variable?

To clarify, we have undertaken logistic regression on a binary outcome (gSG6 IgG seropositivity), which is non-normal (binomial) and non-linear (logit). Therefore, interpretation of exponentiated coefficients is such that a relative percent (fold) increase in outcome is attributable to a relative percent (fold) increase in exposure. While the statistical methodology is outlined in detail in the Appendix 1 (see Page 40-45) we have added the following in the text throughout to clarify the use of the logit link to aid interpretation of model framework (i.e. generalised linear multilevel modelling (*mixed-effects, logistic*)).

(11) Figure 2 – a lot of the "action" is down below 1 HBR. This is expected, and I have no problem with the shape of the fit, but I would be interested in seeing this more clearly in a log-transformed x axis.

Noted, as outlined in our response to Reviewer 1’s Comment #3, we have plotted Figure 2 on the log base 2 scale, but have given equivalent HBR values as labels to retain interpretability and clinical relevance.

(12) Having struggled with the non-linearity up to Figure 3, I couldn't then cope with the increase Figure 3 was communicating. Isn't a percent increase just a linear factor away from an Odds Ratio?

As the associations between HBR/EIR and log odds of gSG6 IgG seropositivity is non-linear, we wanted to show the estimated odds ratio (x-axis) for varying relative per cent increases in the exposure (y-axis), as explained Page 14, Lines 203-205. However, we appreciate this may distract from the main messaging and have moved the HBR and EIR panels of (Previous) Figure 3 in the main manuscript to Figure 2 – Supplement 1 and Figure 4 – Supplement 1 respectively.

(13) For figures 2 and 5 there is obviously a lot of variation left unexplained judging by the scatter – including some quite large studies. I think this should be commented on in the results and discussion. Did the authors check if the risk of bias tended to be linked with outlying observations?

We assessed the risk of bias of outlying observations, and have now added to the manuscript:

(Methods, Page 6-7, Line 145-149) “In order to explore the presence of study-level influence in (HBR and EIR) effect estimate modelling, the Generalised Linear Latent and Mixed Models (gllamm) package [22] was used to produce Cooks distance statistics [23] at the study-level from the generalised linear multilevel models. A conservative cut-off threshold for Cooks distance (4/*n*) was used to guide sensitivity analyses, where studies were excluded, in-turn, to assess outlier influence.”

(Results, Page 22, Line 324-327) **“**Sensitivity analyses exploring potential study-level outlier influence on the estimated associations between anti-gSG6 IgG seroprevalence, HBR and EIR showed no evidence of bias (effect estimates for each sensitivity analysis were consistent with model estimates overall) for studies identified as exhibiting potential influence (HBR: n=6; EIR: n=6).”

(14) I felt uncomfortable in combining Asia and Africa

As outlined above to Comment #7, we chose to directly explore effect-modification of the association between gSG6 IgG seroprevalence and HBR by dominant vector species, rather than by region.

(15) I thought the discussion was well balanced overall. As above, some discussion on the very wide scatter would be useful, and also the fact that the relationship between seroprevalence and HBR is flat above an HBR of 1.5 limits the usefulness of this method in that dynamic range. The authors could comment on how many malaria endemic regions would be covered in the 0-1.5 range. The authors could also comment on the advantage or complimentarity of malaria antigen serology for surveillance in comparison.

We have included a discussion of the diminishing impact of HBR>2 on gSG6 IgG seropositivity and the complementarity of this approach to malarial serosurveillance programs.

(Discussion, Page 22, Line 339-344) “*An. gambiae* gSG6 IgG seropositivity increased with increasing HBR, although these increases had diminishing impact on *An. gambiae* gSG6 IgG seropositivity at higher levels of HBR (approximately greater than 2 bites per person per night). In our study, 17 studies performed across Africa (Angola, Benin, Burkina Faso, Cote d’Ivoire, Senegal) and the Asia Pacific (Cambodia, Myanmar, and the Solomon Islands) reported an HBR <2 demonstrating that the applicability of gSG6 as a biomarker of HBR across a broad range of malaria endemic regions.”

(Discussion, Page 23, Line 373-382) Indeed, positive associations between antibodies specific for *Plasmodium* spp. pre-erythrocytic and blood-stage antigens with gSG6 were demonstrated in analyses of data from diverse malaria endemic areas. […] In these areas, parasite prevalence may therefore overestimate ongoing malaria transmission, making vector surveillance tools essential to informing elimination strategies in the Asia Pacific and other regions where *P. vivax* is endemic.”

References:

1. Sinka ME, Bangs MJ, Manguin S, Rubio-Palis Y, Chareonviriyaphap T, Coetzee M, Mbogo CM, Hemingway J, Patil AP, Temperley WH. A global map of dominant malaria vectors. Parasit Vectors. 2012;5(1):69.

2. Bhatt S, Weiss DJ, Cameron E, Bisanzio D, Mappin B, Dalrymple U, Battle K, Moyes CL, Henry A, Eckhoff PA, Wenger EA, Briët O, Penny MA, Smith TA, Bennett A, Yukich J, Eisele TP, Griffin JT, Fergus CA, Lynch M, Lindgren F, Cohen JM, Murray CLJ, Smith DL, Hay SI, Cibulskis RE, Gething PW. The effect of malaria control on *Plasmodium falciparum* in Africa between 2000 and 2015. Nature. 2015;526(7572):207-11. Epub 2015/09/16. doi: 10.1038/nature15535. PubMed PMID: 26375008.